# The role of actin protrusion dynamics in cell migration through a degradable viscoelastic extracellular matrix: Insights from a computational model

**Tommy Heck**[1], **Diego A. Vargas**[1], **Bart Smeets**[2], **Herman Ramon**[2], **Paul Van Liedekerke**[3,4‡], **Hans Van Oosterwyck**[1,5‡]*

**1** Biomechanics Section, KU Leuven, Leuven, Belgium, **2** MeBioS, KU Leuven, Leuven, Belgium, **3** INRIA de Paris and Sorbonne Universités UPMC Univ paris 6, LJLL Team Mamba, Paris, France, **4** IfADo - Leibniz Research Centre for Working Environment and Human Factors, Dortmund, Germany, **5** Prometheus, Division of Skeletal Tissue Engineering, KU Leuven, Leuven, Belgium

‡ These authors share last authorship on this work.
* hans.vanoosterwyck@kuleuven.be

## Abstract

Actin protrusion dynamics plays an important role in the regulation of three-dimensional (3D) cell migration. Cells form protrusions that adhere to the surrounding extracellular matrix (ECM), mechanically probe the ECM and contract in order to displace the cell body. This results in cell migration that can be directed by the mechanical anisotropy of the ECM. However, the subcellular processes that regulate protrusion dynamics in 3D cell migration are difficult to investigate experimentally and therefore not well understood. Here, we present a computational model of cell migration through a degradable viscoelastic ECM. This model is a 2D representation of 3D cell migration. The cell is modeled as an active deformable object that captures the viscoelastic behavior of the actin cortex and the subcellular processes underlying 3D cell migration. The ECM is regarded as a viscoelastic material, with or without anisotropy due to fibrillar strain stiffening, and modeled by means of the meshless Lagrangian smoothed particle hydrodynamics (SPH) method. ECM degradation is captured by local fluidization of the material and permits cell migration through the ECM. We demonstrate that changes in ECM stiffness and cell strength affect cell migration and are accompanied by changes in number, lifetime and length of protrusions. Interestingly, directly changing the total protrusion number or the average lifetime or length of protrusions does not affect cell migration. A stochastic variability in protrusion lifetime proves to be enough to explain differences in cell migration velocity. Force-dependent adhesion disassembly does not result in faster migration, but can make migration more efficient. We also demonstrate that when a number of simultaneous protrusions is enforced, the optimal number of simultaneous protrusions is one or two, depending on ECM anisotropy. Together, the model provides non-trivial new insights in the role of protrusions in 3D cell migration and can be a valuable contribution to increase the understanding of 3D cell migration mechanics.

**Data Availability Statement:** All data files are available from the Figshare database (https://doi.org/10.6084/m9.figshare.8787119.v1).

**Funding:** The authors thank FWO-Vlaanderen (https://www.fwo.be/) for the research grant G0821.13. T.H. is a Ph.D. fellow of FWO-Vlaanderen. D.A.V. is a postdoctoral fellow, supported by the FWO and EU's Horizon 2020 research and innovation programme (Marie Skłodowska-Curie Grant Agreement No. 665501) (https://ec.europa.eu/programmes/horizon2020/). H.V.O. acknowledges that the research leading to these results has received funding from the European Research Council under the European Union's Seventh Framework Programme (FP7/2007–2013)/ ERC Grant Agreement No. 308223) (https://ec.europa.eu/research/fp7/index_en.cfm). P.V.L would like to acknowledge the support by BMBF -LiSym (https://www.bmbf.de/), INSERM – PhysiCancer (https://www.inserm.fr/), ITMO – INVADE (https://itcancer.aviesan.fr/), EU 7th Framework Programme (NOTOX) (https://ec.europa.eu/research/fp7/index_en.cfm) and ANR – iLite(). The funders had no role in study design, data collection and analysis, decision to publish, or preparation of the manuscript.

**Competing interests:** The authors have declared that no competing interests exist.

## Author summary

The ability of cells to migrate through a tissue in the human body is vital for many processes such as tissue development, growth and regeneration. At the same time, abnormal cell migration is also playing an important role in many diseases such as cancer. If we want to be able to explain the origin of these abnormalities and develop new treatment strategies, we have to understand how cells are able to regulate their migration. Since it is challenging to investigate cell migration through a biological tissue in experiments, computational modeling can provide a valuable contribution. We have developed a computational model of cell migration through a deformable and degradable material that describes both mechanics of the cell and the surrounding material and subcellular processes underlying cell migration. This model captures the formation of long and thin protrusions that adhere to the surrounding material and that pull the cell forward. It provides new non-trivial insights in the role of these protrusions in cell migration and the regulation of protrusion dynamics by cell strength and anisotropic mechanical properties of the surrounding material. Therefore, we believe that this model can be a valuable tool to further improve the understanding of cell migration.

## Introduction

Cell migration is vital for many processes in the human body such as tissue development, wound healing and angiogenesis. In order to migrate, cells adhere to the extracellular matrix (ECM), generate protrusive and contractile forces and degrade the ECM where necessary. These cellular processes are highly affected and regulated by the surrounding ECM, which allows cells to migrate up chemical gradients (chemotaxis), stiffness gradients (durotaxis) and adhesion ligand gradients (haptotaxis) [1]. Cell migration has been studied extensively on 2D substrates as this reduces the complexity of the visualization of cellular processes and the calculation of traction forces applied to the substrate. Cells adhere to and spread on a 2D substrate which gives them a flat shape. They migrate by membrane extension through actin polymerization in wide and flat structures called lamellipodia, followed by adhesion to the substrate at focal adhesion sites, contraction of the cell body by actin stress fibers and retraction of focal adhesions at the rear [2]. However, the physical environment for most cells is three-dimensional (3D) which affects both the shape and migration modes of cells. While cell migration on 2D substrates is well characterized, the subcellular processes underlying 3D cell migration and their dependency on the physical properties of the ECM are less understood. Reported cell migration modes range from bleb-based to protrusion-based [3]. In the former, intracellular pressure results in membrane expulsions called blebs that form without actin polymerization [4]. Cells use these blebs to squeeze through existing pores in the ECM. This fast migration mode, which is characterized by weak cell-ECM adhesion and low ECM degradation, is called amoeboid migration. In the latter mode, actin polymerization results in the formation of actin-rich protrusions that adhere to the ECM through focal adhesions. Actomyosin contraction results in movement of the cell body in the direction of formed protrusions, similar to lamellipodia-driven migration on 2D substrates [5]. The formation and contraction of these protrusions, in combination with strong cell-ECM adhesion and ECM degradation, results in a migration mode that is called mesenchymal migration. Various protrusions have been reported in both migration modes that range in size, function and protrusion mechanism [6–8]. Filopodia are short needle-like protrusions that are used to sense local chemical and mechanical cues, but are not involved displacement of the cell body. Pseudopodia are longer

cylindrical protrusions that form by actin polymerization, similar to lamellipodia in 2D, and are able to displace the cell body by actomyosin contraction. Lobopodia are also longer cylindrical protrusions, but like blebs they are formed by membrane extension due to intracellular pressure. Finally, invadopodia are actin-rich protrusions that have been reported to be used by cancer cells to breach and invade the basement membrane [9]. Here, we focus on 3D migration by formation and contraction of long, actin-rich protrusions. While the migration mode of actin protrusion-driven cell migration has been described, it remains unclear how cells regulate their protrusion dynamics (*e.g.* number of protrusions and protrusion length, lifetime and contractile strength) and what the role is of protrusion dynamics in achieving efficient cell migration that can adapt to the surrounding ECM [8, 10].

In order to migrate cells apply forces to their surrounding ECM by actomyosin contraction. Cells are able to adjust these contractile forces to the local ECM stiffness by a process called mechanosensing [10]. Wolfenson *et al.* demonstrated that fibroblasts cultured on fibronectin-coated elastomeric pillars moved opposing pillars towards each other by actomyosin-based contraction with a constant number of displacement steps per second, resulting in a contraction velocity of 2.5–3.5 nms$^{-1}$ [11]. The contraction lasted until a force of approximately 20 pN was reached, after which contraction was paused for 1–2 s, possibly due to an increased myosin-actin stability at high loads. Once the threshold force was reached, recruitment of $\alpha$-actinin indicated adhesion reinforcement. This could be caused by conformational changes of adhesion molecules upon mechanical stretching as has been observed before for vinculin binding to talin [12]. After the pause the contractile force on the adhesions increased until a higher threshold force was reached, indicating that also the actomyosin contraction reinforces. The number of contraction steps required to reach a threshold force was highly dependent on ECM rigidity. On soft pillars more steps are required to build up the force than on stiff pillars as the soft pillars deflect more. Therefore, on stiff pillars the threshold force is reached more frequent and as a consequence a higher actomyosin contractile force and a stronger focal adhesion are obtained. This results in the same displacement of the pillars independent of their stiffness. A similar effect has been observed for epithelial cells on a micropillar substrate [13]. In this way cells are able to generate more force in the direction of protrusions that sense a higher local ECM stiffness. However, it is not clear how this affects 3D migration of cells that extend multiple actin protrusions at the same time.

Computational models have been developed to unravel the mechanisms underlying 3D cell migration. Kim *et al.* developed a model of cell invasion into a discrete fibrillar ECM [14]. The main emphasis of their work, however, does not lie on 3D migration mechanisms, but on the development of a method to characterize the local stiffness sensed by a cell in a fibrillar material. The cell, which is modeled as an active deformable object, extends small finger-like protrusions called filopodia that probe the local ECM stiffness. The polarization direction of the cell rotates towards the direction of highest ECM stiffness and defines filopodia lifetime and the direction of lamellipodium protrusion. This model provides great insight in the way cells can orient themselves towards the direction of higher ECM stiffness by mechanosensing of the local environment. However, cell migration in this model is not the result of contraction of long and thin protrusions to displace the cell body, as has been reported for 3D cell migration. Instead, migration is the result of lamellipodium protrusion and actomyosin contraction at the cell body. Moure *et al.* developed a model for 2D and 3D, spontaneous and chemotactic amoeboid migration [15–17]. They used the phase-field method to track the cell and their model captures myosin and globular and filamentous actin. The cell migrates by expanding and retracting pseudopods. Pseudopods expand by local actin protrusion that generates an outward stress to the membrane. Their dynamics (growth time, time interval of initiation and location of initiation) is regulated by probability functions derived from experiments. In the

case of chemotactic migration the probability of pseudopod initiation is modified according to the average chemoattractant gradient at the cell membrane. The model is able to simulate realistic cell shape dynamics and migration paths for amoeboid migration through a fibrous environment, with fibers modeled as rigid obstacles. However, it does not capture a degradable and deformable ECM yet for the cell to migrate through. Zhu *et al*. modeled both the cell and ECM as a collection of nodes and springs [18]. By varying processes as actin protrusion, actomyosin contraction, cell-ECM adhesion and ECM degradation they were able to obtain six experimentally described migration modes including mesenchymal and blebbing. Their mesenchymal migration mode is characterized by membrane expansion at the front due to actin polymerization and retraction of the rear due to actomyosin contraction. However, it does not capture the formation, mechanosensing and contraction of multiple competing thin protrusions. Ribeiro *et al*., and in a follow up study Merino-Casallo *et al*., captured this protrusion competition in their mechanical cell model by extending protrusions represented by vectors attached to a central connection point [19, 20]. Protrusion growth and retraction are regulated by chemosensing and constrained by the ECM. The cell migrates by retraction of the longest protrusion and the model is able to perform chemotaxis. However, the model does not capture contact of the cell body with the solid ECM and does not include mechanosensing.

Here, we present a computational model of cell migration through a degradable viscoelastic ECM. This model is a 2D representation of 3D cell migration. The cell is modeled as an active deformable object that captures the viscoelastic behavior of the actin cortex and the subcellular processes underlying 3D cell migration. The ECM is regarded as a viscoelastic material and is modeled by means of the smoothed particle hydrodynamics (SPH) method. Compared to the models described above, our model both describes the mechanics of the actin cortex and the ECM and captures the formation of competing protrusions that adhere to the ECM, probe the local ECM and contract in order to displace the cell body. We use this model to investigate the role of protrusion dynamics and ECM mechanics on cell migration. We demonstrate that changes in ECM stiffness and cell strength affect cell migration and are accompanied by changes in protrusion dynamics (*i.e.* total number of protrusions, protrusion length and protrusion lifetime), while directly changing protrusion dynamics does not affect cell migration. A stochastic variability of protrusion lifetimes is enough to regulate cell migration. Force-dependent adhesion disassembly can increase the efficiency of cell migration by reducing the number of protrusions, but does not result in faster migration. We also demonstrate that the optimal number of simultaneous protrusions for cell migration is one or two, depending on the anisotropy of the ECM.

## Methods

Our modeling strategy is a hybrid approach in which the individual cell is represented by an agent-based model and the degradable viscoelastic ECM by a meshless Lagrangian particle-based method (see Fig 1A). The cell is modeled as an active 2D deformable object (see [21, 22] as examples of deformable cell models), for which the boundary is discretized by viscoelastic elements, that represents the viscoelastic behavior of the membrane and underlying actin cortex and the interaction with the ECM. The cell model captures the subcellular processes underlying cell migration, *i.e.* protrusion formation, cell-ECM adhesion, ECM mechanics-regulated actomyosin contraction and ECM degradation.

The ECM is regarded as a 2D continuous viscoelastic material that represents nanoporous non-fibrillar hydrogels such as polysaccharide based gels (agarose, alginate) or synthetic gels (polyethylene glycol (PEG)). A 2D planar cross section is considered in order to reduce computational cost and complexity. Strain stiffening of the material is added in some

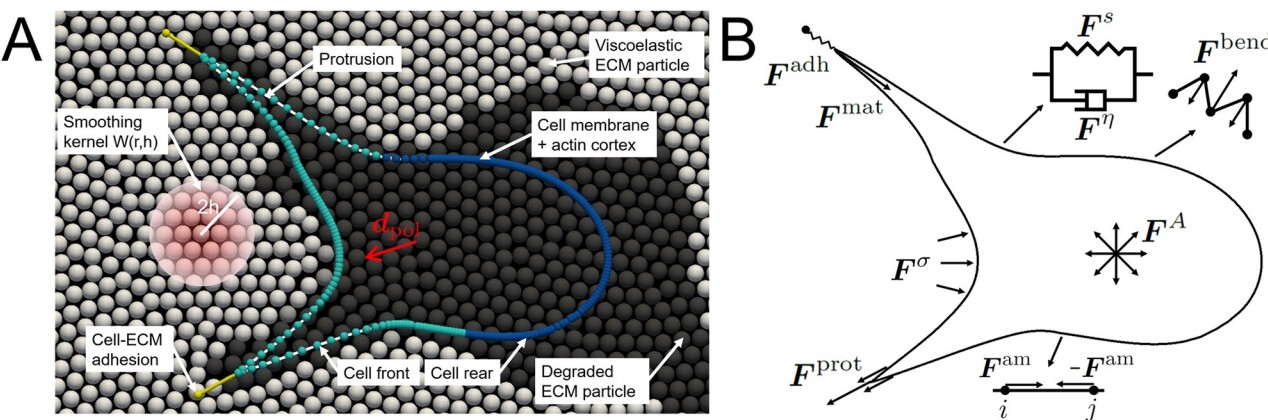

**Fig 1. Cell migration model overview.** (A) Overview of the model of cell migration through a degradable viscoelastic matrix. The cell forms protrusions at the cell front that degrade the ECM particles and adhere to the ECM. The cell polarization direction rotates towards existing adhesion directions and defines the cell front (light blue particles) and rear (dark blue particles). Viscoelastic ECM particles (white) are degraded gradually by fluidization from partially degraded particles (gray) to fully degraded ECM particles (black). The ECM is modeled as a continuous material by using a smoothing kernel. (B) Schematic overview of the mechanical representation of the actin cortex. The following forces are indicated: cortex elastic force $F^s$, cortex viscous force $F^\eta$, cortex bending force $F^{\text{bend}}$, area conservation force $F^A$, cortex-ECM elastic contact force $F^\sigma$, protrusion force $F^{\text{prot}}$, adhesion force $F^{\text{adh}}$, actomyosin contractile force $F^{\text{am}}$ and maturation force $F^{\text{mat}}$ (see also Eq 1).

simulations to model the nonlinear and anisotropic mechanical behavior of fibrillar hydrogels like collagen gels. The ECM is modeled by means of the meshless Lagrangian SPH method. In this method, a continuous material is discretized into elements, called particles, for which material properties and variables (*e.g.* mass, density, velocity and hydrostatic pressure) are computed. The use of a smoothing kernel allows to discretize the continuum laws of fluid and solid mechanics. As discussed before, the meshless character of SPH allows to naturally capture discrete processes in a continuum material [23]. Besides, meshless methods can deal with deformable interfaces [24], large deformations and discontinuities [23]. In the next sections the implementation of cell and ECM mechanics and protrusion dynamics is described. An overview of the model parameters is given in Table 1.

## Deformable cell model

The cell model consists of viscoelastic elements (particles connected by a line segment representing an elastic spring and a viscous damper in parallel) that capture the viscoelastic behavior of the actin cortex underlying the cell membrane. As cells migrate in a low Reynolds number environment, inertial forces can be neglected. Therefore, the conservation of momentum equation for a boundary particle $i$ of the cell reads:

$$\sum_j \eta_c(\hat{\boldsymbol{e}}_{ij} \cdot \boldsymbol{v}_{ij})\hat{\boldsymbol{e}}_{ij} + \sum_k \Gamma_{ik}\boldsymbol{v}_{ik} + \gamma_{\text{liquid}}\boldsymbol{v}_i$$
$$= \boldsymbol{F}_i^s + \boldsymbol{F}_i^{\text{bend}} + \boldsymbol{F}_i^A + \boldsymbol{F}_i^{\text{rep}} + \boldsymbol{F}_i^\sigma + \boldsymbol{F}_i^{\text{prot}} + \boldsymbol{F}_i^{\text{adh}} + \boldsymbol{F}_i^{\text{am}} + \boldsymbol{F}_i^{\text{mat}},$$

(1)

with on the left-hand side the velocity-dependent terms (drag forces) and on the right-hand side all the forces that work on the cell boundary (see Fig 1B). The passive cell mechanics is modeled with an actin cortex elastic spring force $F^s$, a cortex bending rigidity force $F^{\text{bend}}$, a cell area conservation force $F^A$ and a repulsive Hertz-like force $F^{\text{rep}}$. In order to allow the formation of long and sharp protrusions, cortex bending rigidity ($F^{\text{bend}}$) and cell area conservation ($F^A$) are assumed to be weak and applied only to prevent membrane folding and cell shrinking. The repulsive force ($F^{\text{rep}}$) is applied to cell particles that approach a line segment in order to

**Table 1. Model parameters.**

| | Parameter | Symbol | Value | Units | Ref |
|---|---|---|---|---|---|
| **Cell** | radius | $R$ | 15 | μm | [5] |
| | initial spring rest length | $l_0$ | 0.4 | μm | model setup[2] |
| | cortex stiffness | $k_s$ | $2.8\times10^{-3}$ | N/m | [25] |
| | cortex viscosity | $\eta_c$ | $2.5\times10^{-3}$ | Ns/m | [14, 21, 26] |
| | bending rigidity constant | $k_{\text{bend}}$ | $2\times10^{-15}$ | Nm | [25] |
| | area constraint constant | $k_A$ | 0.71 | N/m$^2$ | trial runs |
| | repulsive force constant | $k_{\text{rep}}$ | 1.4 | N/m$^{1.5}$ | trial runs |
| | repulsive force threshold distance | $d_{\text{rep}}$ | 0.2 | μm | trial runs |
| | liquid drag force constant | $\gamma_{\text{liquid}}$ | 40 | Pa $\cdot$ s | trial runs |
| | polarization rate | $r_{\text{pol}}$ | $1.1\times10^{-3}$ | s$^{-1}$ | [14, 27] |
| **Protrusion** | protrusion formation rate | $r_{\text{prot}}$ | $6\times10^{-5}$ | s$^{-1}$ | [28][1] |
| | protrusion particle width | $w_{\text{prot}}$ | 20 | | model setup[2] |
| | protrusion force particle width | $w_f$ | 3 | | model setup[2] |
| | protrusion cortex stiffness | $k_{s,\text{prot}}$ | $2.8\times10^{-6}$ | N/m | trial runs |
| | protrusion force | $F_{\text{prot}}$ | 0.32 | nN | [28, 29][1] |
| | protrusion time | $T_{\text{prot}}$ | 400 | s | [28, 29][1] |
| | protrusion deflection rate | $r_{\text{defl}}$ | 0.1 | s$^{-1}$ | trial runs |
| | protrusion finish time | $T_{\text{finish}}$ | 400 | s | trial runs |
| **Adhesion** | adhesion stiffness | $k_{\text{ad}}$ | $2\times10^{-3}$ | N/m | trial runs |
| | initial adhesion length | $l_{0,\text{adh}}$ | 5 | μm | model setup[2] |
| | minimal disassembly rate | $r_{\text{off,min}}$ | $2.78\times10^{-4}$ | s$^{-1}$ | [28, 29][1] |
| | zero force disassembly rate | $r_{\text{off,0}}$ | 0.2 | s$^{-1}$ | [30][1] |
| | force-dependent disassemble parameter | $\zeta_{\text{diss}}$ | $2\times10^4$ | | [30][1] |
| | adhesion rupture constant | $f_{\text{rupt}}$ | 1.0 | | |
| **Maturation and contraction** | reference actomyosin force | $F_{\text{am}}$ | 0.6 | nN | [31, 32] |
| | maturation time | $T_{\text{mat}}$ | 600 | s | trial runs |
| | optimal cortex curvature | $\kappa_0$ | −0.15 | | [33] |
| | myosin II-binding curvature range | $\kappa_w$ | 0.25 | | [33] |
| **ECM degradation** | degradation rate at protrusion tip | $r_{\text{degr,tip}}$ | 0.2 | s$^{-1}$ | trial runs |
| | degradation distance at protrusion tip | $d_{\text{degr,tip}}$ | 2.4 | μm | model setup[2] |
| | degradation rate at protrusion | $r_{\text{degr,prot}}$ | 0.033 | s$^{-1}$ | trial runs |
| | degradation distance at protrusion | $d_{\text{degr,prot}}$ | 1.4 | μm | model setup[2] |
| | degradation rate at cell body | $r_{\text{degr,cell}}$ | 0.033 | s$^{-1}$ | trial runs |
| | degradation distance at cell body | $d_{\text{degr,cell}}$ | 1.9 | μm | model setup[2] |
| | solid hydrostatic pressure threshold | $p_{\text{th,degr}}$ | 20 | Pa | [34, 35][1] |
| **ECM SPH** | particle distance | $dp$ | 2.0 | μm | |
| | smoothing length | $h$ | 2.6 | μm | |
| | initial density | $\rho_0$ | 1000 | kg/m$^3$ | |
| **Homogeneous ECM** | Young's modulus | $E_{\text{ECM}}$ | 200 | Pa | |
| | Poisson's ratio | $v$ | 0.45 | | |
| | dynamic viscosity | $\mu$ | 1000 | Pa $\cdot$ s | |
| **Strain stiffening** | linear stiffness fibers | $k_0$ | $1\times10^3$ | N/m | [36] |
| | strain stiffening onset strain | $\epsilon_s$ | 0.075 | | [36] |
| | exponential strain stiffening constant | $d_s$ | 0.033 | | [36] |

[1] Parameters are fitted to mimic protrusion, adhesion and contraction dynamics observed in experiments.

[2] Parameters are selected based on the cell and ECM resolution.

prevent the cell boundary from penetrating itself. A more detailed description of the cell model mechanics can be found in S1 Text. The cell is embedded in the ECM and has physical interaction with solid particles of the ECM, which is captured by $F^\sigma$. The remaining forces capture membrane protrusion by actin polymerization ($F^{\text{prot}}$), cell-ECM adhesion ($F^{\text{adh}}$), actomyosin contraction ($F^{\text{am}}$) and mechanosensing-regulated protrusion maturation ($F^{\text{mat}}$). These processes are described in more detail in the following sections.

The left-hand side describes dissipation of the actin cortex, with the actin cortex friction $\eta_c$, the velocity $v$ and the normal unit vector from particle $j$ to $i$ $\hat{e}_{ij}$, for connected cell boundary particles $j$ (the notation $v_{ij} = v_i - v_j$ will be used for all vectors later on), viscous cell-ECM forces for contact with neighboring ECM particles $k$ (see S2 Text) and a drag force $\gamma_{\text{liquid}} v_i$ due to interaction with the culture medium. The cell locally degrades the ECM by fluidization of solid ECM particles. By permitting these fluid particles to move through the cell boundary, the cell is allowed to migrate through the ECM. The cell model initially has a circular shape with a radius of 15 μm and consists of 235 particles connected by line segments, with a particle distance of 0.4 μm.

## Extracellular matrix model

The ECM is modeled as a continuous degradable viscoelastic material by the SPH method. In this method a material is divided into a set of discrete elements, called particles, for which material properties (*e.g.* mass, density, velocity and stress) are described. A Wendland smoothing kernel [37] $W(r, h)$ (see Fig 1A), with $r$ the distance to a neighboring particle and $h$ the smoothing length, is used to approximate these properties and to implement the laws of fluid and solid mechanics in a discrete manner. Again, as cellular processes (μm-scale) occur at a low Reynolds number, viscous forces will dominate over inertial forces leading to an overdamped system. Therefore, inertial forces can be omitted from the conservation of momentum equation, resulting in the non-inertial SPH (NSPH) method. As described before [23, 38], the conservation of momentum for ECM particle $i$ in contact with neighboring particles $j$ becomes:

$$-m_i \sum_j m_j \frac{\mu_i + \mu_j}{\rho_i \rho_j} \frac{x_{ij} \cdot \nabla_i W_{ij}}{|x_{ij}|^2 + \eta^2} v_{ij} = m_i \sum_j m_j \left( \frac{\sigma_i}{\rho_i^2} + \frac{\sigma_j}{\rho_j^2} \right) \cdot \nabla_i W_{ij} + F_i^b, \qquad (2)$$

with $m$ the mass, $\rho$ the density, $\mu$ the dynamic viscosity, $v$ the velocity, $x$ the position, $\sigma$ the stress tensor, $\nabla_i W_{ij}$ the derivative of the smoothing kernel $W$, $\eta = 0.01h^2$ a correction factor that prevents singularity when particles approach each other and $F_i^b$ body forces. The detailed implementation of this method as described before [23, 38] is summarized in S2 Text. The ECM is modeled as a circular domain with a radius of 150 μm, fixed displacement at the boundary and a particle distance $dp = 2$ μm. It is modeled as a viscoelastic material with a Young's modulus $E_{\text{ECM}} = 200$ Pa, Poisson's ratio $v = 0.45$ and dynamic viscosity $\mu = 1000$ Pa · s.

*In vivo* ECMs contain fibrillar proteins like collagen that induce nonlinear and anisotropic mechanical properties. Strain stiffening of the material by collagen is captured in some simulations (see section Optimal number of simultaneous protrusions depends on ECM anisotropy) by placing nonlinear elastic springs between ECM particles (see Fig 2A and 2B). These springs do not embody individual collagen fibers, but are a coarse-grained representation of the nonlinear mechanical material behavior. Therefore, the mechanics of a fibrillar ECM is captured, but structural properties such as individual fibers and pores are not included. We note that alternatively, a similar nonlinear mechanical behavior of the ECM could in principle be captured by assuming a strain-dependent Young's modulus in the SPH model, but we did not

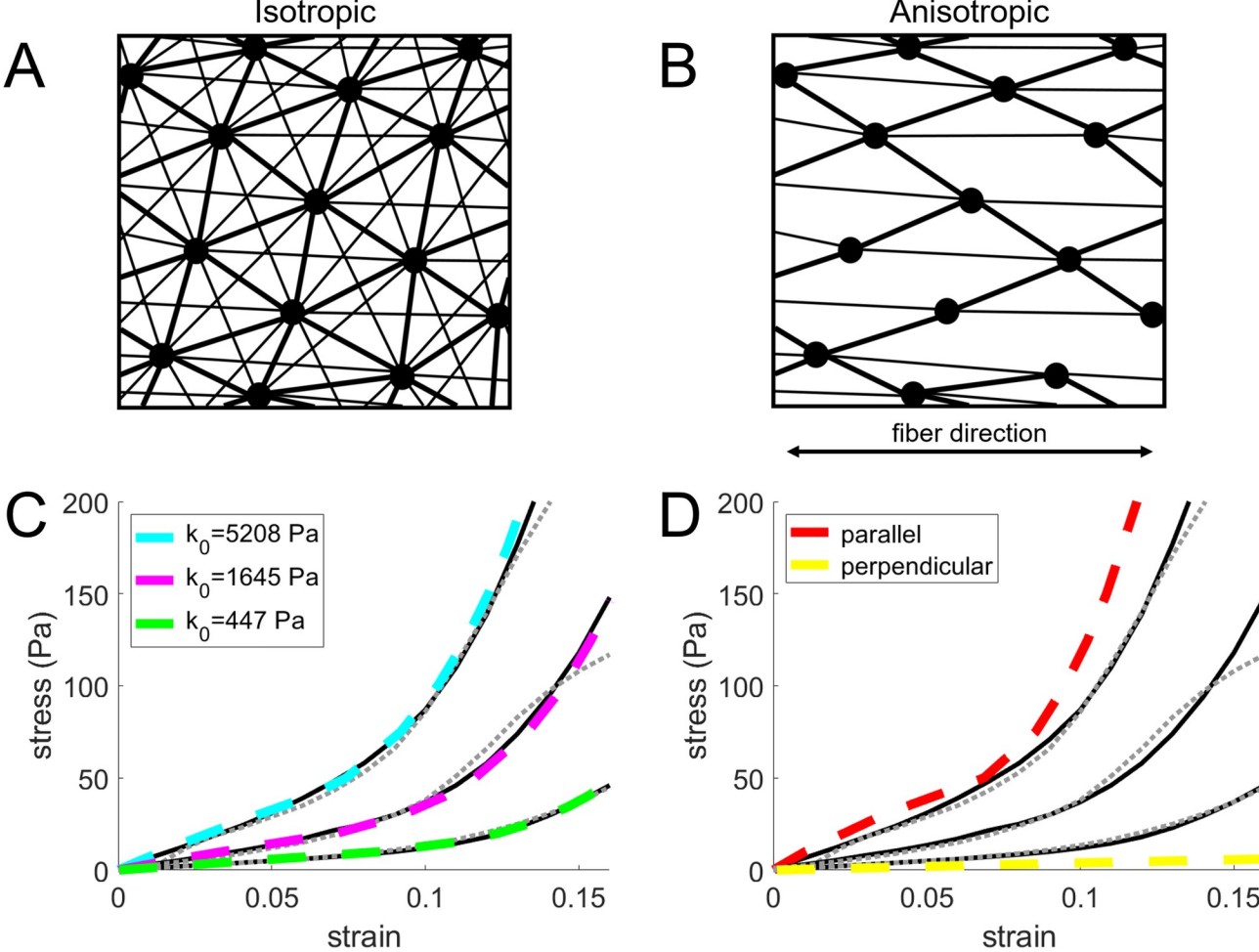

**Fig 2. SPH isotropic and anisotropic fibrillar ECM model validation.** (A) Illustration of springs (lines) between ECM particles (dots) for an isotropic fibrillar ECM and (B) an anisotropic, uniaxial fibrillar ECM. The line thickness emphasizes the weighted contribution of springs based on particle distance (see Eqs 3 and 4). (C) Stress-strain curves for uniaxial stretching of hydrogels with three different collagen concentrations (0.6, 1.2, and 2.4 mg/ml, from shallowest to steepest curves). Dashed gray lines indicate extensional rheometer measurements and black solid lines indicate finite-element model fit, both recreated from data from Steinwachs *et al.* [36]. Light blue, magenta and green dashed lines show the results obtained for the fibrillar SPH model with identical material parameters ($k_0 = 447$, 1645 or 5208 Pa for the 3 collagen concentrations, $\epsilon_s = 0.075$ and $d_s = 0.033$). (D) Red and yellow dashed lines show the results obtained for the anisotropic, uniaxial fibrillar SPH model stretched along the fiber direction (parallel) or perpendicular to the fiber direction.

pursue this option. The implementation used here is based on a study performed by Steinwachs *et al.* in which the nonlinear stress-strain relationship of collagen due to fiber stiffening and buckling is captured in a constitutive equation that describes the mechanical behavior of the bulk material [36]. Here, strain stiffening is implemented by adding nonlinear springs between ECM particles. The total fiber spring force $\boldsymbol{F}_i^{\text{fib}}$ applied on particle $i$ from springs connected to neighboring particles $j$ is:

$$\boldsymbol{F}_i^{\text{fib}} = \sum_{j \in \mathcal{S} \setminus i} - w_{ij}(f_{\text{degr},i}, f_{\text{degr},j}) k_{\text{fib},ij} (\boldsymbol{x}_{ij} - \boldsymbol{x}_{0,ij}), \tag{3}$$

with $\mathcal{S}$ the set of solid ECM particles (see S3 Text), $\boldsymbol{x}_0$ the initial particle position, $k_{\text{fib},ij}$ a strain-dependent spring stiffness and $w_{ij}$ a factor that weighs the contribution of each spring based

on the particle distance and local kernel support:

$$w_{ij} = \frac{1}{2}\left( \frac{\frac{m_i}{\rho_i}W_{ij}}{\sum\limits_{k\in\mathcal{S}\backslash j}\frac{m_k}{\rho_k}W_{jk}} + \frac{\frac{m_j}{\rho_j}W_{ij}}{\sum\limits_{k\in\mathcal{S}\backslash i}\frac{m_k}{\rho_k}W_{ik}} \right).$$

(4)

The spring stiffness $k_{\text{fib},ij}$ depends on the strain $\epsilon_{ij}$ between particles $i$ and $j$ as described in [36], but with ignoring fiber buckling:

$$k_{\text{fib}}\left(\epsilon_{ij}\right) = \begin{cases} 0 & \text{for } \epsilon_{ij} \leq 0 \\ k_0 & \text{for } 0 < \epsilon_{ij} \leq \epsilon_s \\ k_0 e^{\frac{(\epsilon_{ij} - \epsilon_s)}{d_s}} & \text{for } \epsilon_{ij} > \epsilon_s \end{cases},$$

(5)

with $\epsilon$ the strain, $k_0$ the linear stiffness, $\epsilon_s$ the strain threshold for the onset of strain stiffening and $d_s$ an exponential strain stiffening coefficient. Compared to the model of Steinwachs *et al.* fiber stiffness is neglected completely under compression. The mechanical behavior of the nonfibrillar matrix is captured with a strongly reduced Young's modulus of 10 Pa. Steinwachs *et al.* obtained values for these model parameters by fitting their finite element model to measurements of uniaxial stretching of collagen hydrogels in an extensional rheometer with different collagen concentrations (0.6, 1.2, and 2.4 mg/m) [36]. Uniaxial stretching simulations are performed with our SPH strain stiffening ECM model with the same parameter values. The results of these simulations are shown in Fig 2C) together with the results obtained in [36].

It can be seen that the stress-strain curves obtained for our model agree very well with those obtained in [36], which indicates that our model is able to capture strain stiffening caused by collagen fibers.

Next, the model described above is adapted in order to model an anisotropic collagen gel with a preferred fiber direction. Strain stiffening springs are placed only between particles for which the angle between a prescribed fiber direction and a vector connecting these two particles is equal to or lower than 30˚ (see Fig 2B). As this strongly reduces the number of springs in the model, the linear stiffness $k_0$ is increased to 10 kPa. The results of simulations of stretching a gel along the fiber direction or perpendicular to the fiber direction are shown in Fig 2D. It can be seen that the gel is slightly stiffer along the fiber direction, but is very soft along the direction perpendicular to the fiber direction. In this way the effect of ECM anisotropy on cell migration can be investigated.

## Protrusion dynamics

The cell migrates through the ECM by forming protrusions that adhere to and probe the local ECM and contract to displace the cell body. This section describes the dynamics of a protrusion during its lifetime. First, locations of membrane protrusion are randomly selected at the cell front, which is defined by a polarization direction of the cell (section Protrusion initiation and cell polarization). Protrusions grow, for a prescribed time, by weakening the actin cortex and pushing the membrane outwards through directed actin polymerization, after which the protrusion adheres to the ECM (section Protrusion growth and cell-ECM adhesion). Next, the protrusion probes the ECM and matures based on the local ECM stiffness, again for a prescribed time (section Protrusion maturation). After maturation, the protrusion contracts with a force that is scaled by the amount of maturation and thereby displaces the cell body in the

adhesion direction (section Actomyosin contraction). In order to migrate the cell needs to locally degrade the ECM (section ECM degradation). Finally, the adhesion disassembles with a force-dependent probability, after which the protrusion retracts and the cortex is available again to form new protrusions (section Adhesion disassembly). Due to the fixed growth and maturation times, differences in lifetime between protrusions are only caused by adhesion disassembly. Competition between protrusions, due to differences in the amount of maturation between protrusions, can then result in migration regulated by ECM properties.

**Protrusion initiation and cell polarization.** Cell boundary particles can be selected to initiate the formation of a protrusion with a chosen rate $r_{prot}$, resulting in an exponential probability distribution of protrusion initiation. As soon as the first protrusion is initiated, the cell is polarized, with polarization direction $\boldsymbol{d}_{pol}$ (see Fig 1A) equal to the protrusion growth direction $\boldsymbol{d}_{prot}$ (the direction from the cell center of mass to the selected protrusion particle at protrusion initiation). This polarization direction is used to define a front, consisting of the first 50% of cell particles along the polarization direction, and a rear of the cell. Protrusions can be initialized only at the front of the cell. As adhesions have been reported to appear as a trademark of polarization [39], $\boldsymbol{d}_{pol}$ targets and rotates towards the average direction of mature adhesions, seen from the center of mass of the cell, with a chosen polarization rate $r_{pol}$ (see S1 Fig). After reaching this target direction, $\boldsymbol{d}_{adh}$ remains unchanged until the average adhesion direction is altered again. It is important to note that, although a single polarization direction is defined, multiple protrusions will be initialized at the front half of the cell and will grow in different directions. Therefore, competition between multiple protrusions, that affect cellular polarization, is captured in this model.

**Protrusion growth and cell-ECM adhesion.** Protrusions form by local weakening of the actin cortex and actin polymerization underneath the membrane that pushes the membrane outwards. A protrusion consists of the selected cell particle and the first 20 particles along both directions of the cell boundary (see Fig 3A). This number of particles is chosen to allow the formation of multiple large protrusions by stretching the cell boundary, while maintaining an adequate boundary resolution relative to the resolution of the ECM. The protrusion is allowed to form only if none of these particles is already part of another protrusion. The stiffness of elastic springs between the protrusion particles is decreased by three orders of magnitude to account for actin cortex weakening [29]. A protrusion force $\boldsymbol{F}^{prot}$, caused by polymerizing actin pushing against the membrane, is applied in the protrusion growth direction to the central 7 particles of the protrusion. This results in a velocity of protrusion growth similar to the ~1.5–5 μm/min reported for cells embedded in collagen [28, 29]. Actin is assumed to polymerize in the protrusion growth direction. Therefore, the protrusion force on a particle is reduced when the angle between the local normal vector to the cell boundary $\hat{\boldsymbol{n}}$ and the protrusion growth direction increases, resulting in a thin and sharp protrusion (see Fig 3B).

In order for the cell to form protrusions and migrate through a continuous ECM, the ECM has to be degraded. This is modeled by fluidization of ECM particles, which is captured by a degradation factor $f_{degr}$ that has a value between 1 (intact solid ECM) and 0 (fully degraded ECM) as introduced before in [23]. ECM particles close to the protrusion tip or the cell body can be degraded with a chosen degradation rate. Contrary to the solid ECM particles, the degraded ECM is not assumed to act as a physical obstacle for the cell as it should be easily displaced through the nanoporous ECM. However, this is not possible for fluid particles in the ECM model. Instead, fluid particles are allowed to move freely through the cell boundary. In this way the cell can form tunnels by ECM degradation through which it can migrate more easily, while full kernel support and buildup of hydrostatic pressure in the ECM are preserved. The implementation of a fluid and solid ECM state requires adaptation of the SPH formulation and cell-ECM boundary conditions, which is described in S3 Text. ECM particles within a

### Protrusion formation

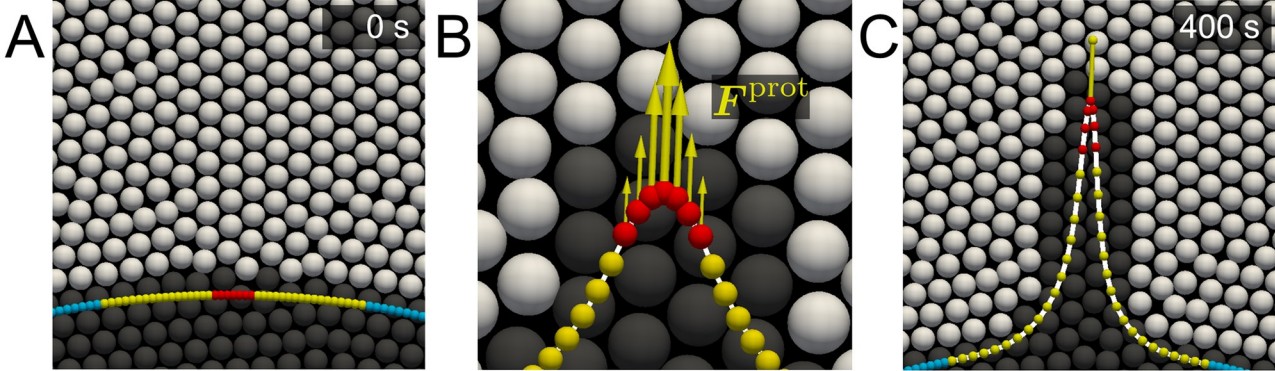

### Protrusion maturation and contraction

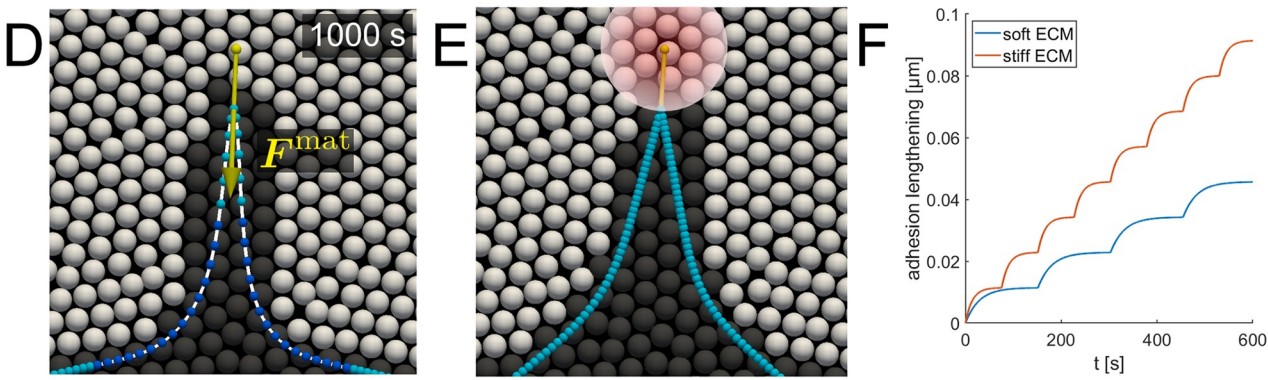

**Fig 3. Protrusion formation, maturation and contraction.** (A) Protrusion particles are selected for which the actin cortex stiffness $k_s$ is lowered (red and yellow) and to which an actin protrusion force $F^{\mathrm{prot}}$ is applied (red). (B) ECM particles in contact with the protrusion force particles are degraded allowing the protrusion to form. (C) At the end of protrusion growth a cell-ECM adhesion is formed at the protrusion tip. (D) Boundary particles at the protrusion base are fixed and a contractile force $F^{\mathrm{mat}}$ is applied to the adhesion boundary particle. (E) After maturation the protrusion contracts and displaces the cell body in the protrusion direction, where the transparent circle indicates the influence area of the adhesion by using the smoothing kernel. (F) During maturation the contractile force is increased every time the adhesion is stretched above a threshold length. As the threshold is reached more frequent on a stiff ECM (red) compared to a soft ECM (blue), a protrusion generates more force in a stiffer ECM.

distance of 2.4 μm (1.2 times the ECM particle distance $dp$) from the protrusion force particles are degraded with a chosen degradation rate $r_{\mathrm{degr,tip}}$, which creates space in the ECM for the protrusion to grow. Degradation of the ECM close to the protrusion tip is only allowed during protrusion growth. Afterwards, ECM degradation only occurs close to the cell body as will be explained in section ECM degradation.

When a protrusion encounters ECM while growing, it should be easier for a protrusion to grow in the direction of least resistance by deflecting away from the ECM. This is accounted for by rotating the protrusion growth direction towards the degraded ECM as described in S4 Text. The protrusion force is applied for $T_{\mathrm{prot}} = 400$ s, resulting in a protrusion length of 25–30 μm which falls within the range of protrusion lengths of 10–78 μm reported in literature [28, 29]. A prescribed growth time is chosen because the factors that regulate protrusion growth, such as chemical signaling, actin availability and mechanochemical feedback at the protrusive front are not included. This also allows to vary the protrusion growth time between cells in order to investigate the effect of protrusion length (which is reasonably equivalent to protrusion growth time) on cell migration After protrusion growth a cell-ECM adhesion is

formed at the protrusion tip. The adhesion is modeled as a spring, with stiffness $k_{ad}$, that connects the protrusion tip with a point in the ECM 5 μm from the protrusion tip in the protrusion growth direction (see Fig 3C). The adhesion is chosen to be significantly stiffer than the ECM such that mechanosensing by the protrusion depends solely on ECM mechanics (see section Protrusion maturation). The smoothing kernel is used to distribute the adhesion force over the neighboring ECM particles and to calculate the displacement of the adhesion point as the ECM deforms. By using a distribution of the adhesion rather than adhering to a single ECM particle the adhesion can bind to an arbitrary point in the continuous ECM and numerical instability due to application of a large force to a single ECM particle is prevented.

**Protrusion maturation.** Cells are able to adjust their contractile force to the local ECM properties by a process called mechanosensing. Here, after the protrusion has formed and adhered to the ECM, the protrusion and adhesion mature according to the mechanosensing mechanism described by Wolfenson *et al.* [11]. An actomyosin contractile force $\boldsymbol{F}^{mat}$, with magnitude equal to a reference actomyosin contractile force $F_{am}$ multiplied by a maturation factor $f_{mat}$ initially set to 0.1 (with maximal contraction if $f_{mat} = 1.0$), is applied to the adhesion boundary particle in the direction opposite to the protrusion growth direction and stretches both the adhesion and the ECM (see Fig 3D). For a period of 600 s, $f_{mat}$ is increased with 0.1 every time the adhesion is stretched above a threshold length, equivalent to 98% of the applied force step of 0.1 times $F_{am}$ (see Fig 3F). In this way, actomyosin is reinforced when actomyosin contraction and adhesion stretching are balanced. This balance is reached faster for a stiff ECM, because a stiff ECM has to be displaced less by the application of $\boldsymbol{F}^{mat}$. Therefore, this mechanism results in more actomyosin reinforcement (higher $f_{mat}$) for a time period of 600 s and thus stronger protrusions in stiff ECMs. A fixed and equal maturation time is prescribed for all protrusions to ensure that differences in protrusion and adhesion maturation are only the result of implemented mechanosensing mechanisms. A maximum of 10 maturation steps, equivalent to $f_{mat}$, is allowed in the model. Therefore, the protrusion strength saturates in very stiff ECMs. It is assumed that during the maturation process the actin cortex in the newly formed protrusion is not yet restored and therefore the force on the adhesion is not affected by contraction of the cell body or any other protrusion. If not, contraction of a neighboring protrusion would affect the stretching of the adhesion, which makes it very challenging to detect the moment of actomyosin reinforcement in the model. Therefore, in order to assure that stretching of the adhesion happens only due to $\boldsymbol{F}^{mat}$ of the corresponding protrusion, the base of the protrusion is fixed during maturation by increasing the liquid drag force constant (see Fig 3D). At the end of the maturation phase $f_{mat}$ is set to 0.1 times 600 s divided by the average time per maturation step, with a maximal value of 1. The linear spring stiffness of the protrusion particles is restored, which represents restoring of the actin cortex under the protrusion membrane. As the actin cortex is assumed to be restored in an unstretched state, the rest length of springs between the protrusion particles is set to the new distance between the protrusion particles after protrusion formation. Finally, the contractile force $\boldsymbol{F}^{mat}$ is removed from the adhesion boundary particle and replaced by contraction of the actin cortex of the entire protrusion, which allows the protrusion to displace the cell body (see Fig 3E).

**Actomyosin contraction.** Cells migrate by displacing their cell body through actomyosin contraction. As Fischer *et al.* showed that F-actin, myosin IIA and myosin IIB colocalize in the cortex and form longitudinal bundles similar to stress fibers in 2D, actomyosin contraction is assumed to occur only in the actin cortex [29]. Actomyosin contraction is applied to both the cell body ($f_{mat} = 0.05$) and mature protrusions and is the main driver of cell displacement. A contractile force $\boldsymbol{F}^{am}$ is applied to the two particles connected by each line segment, with magnitude equal to $F_{am}$ times the lowest $f_{mat}$ of both particles. Therefore, the contractile force on

particle $i$ applied by the neighboring particles $j$ is:

$$\boldsymbol{F}_i^{\mathrm{am}} = \sum_j \min\left(f_{\mathrm{mat},i}, f_{\mathrm{mat},j}\right) f_{\mathrm{curv},ij} F_{\mathrm{am}} \hat{\boldsymbol{e}}_{ji}, \tag{6}$$

with $f_{\mathrm{curv}}$ a cortex curvature factor. Elliott *et al.* demonstrated that myosin II in endothelial cells associates stronger to the actin cortex at regions of low cortex curvature [33]. Myosin II contractility at these regions acts to maintain this minimal curvature, thereby functioning as a positive feedback mechanism that regulates cell shape and protrusion assembly and disassembly. In order to capture this curvature-dependent contraction, the contractile force is scaled by a curvature factor $f_{\mathrm{curv}}$ which has a value between 0 and 1 based on the local cortex curvature $\kappa$. The dependence of $f_{\mathrm{curv}}$ on the local curvature is based on experimental data provided by Elliott *et al.* as explained in S5 Text [33]. In order to allow contraction of protrusions and prevent that the cell boundary keeps growing as more protrusions are formed, the rest length of the elastic actin cortex springs is reduced as the boundary contracts. For all line segments not part of a growing or maturing protrusion, the rest length is set to the current distance between the boundary particles if this distance is smaller than the current rest length and longer or equal to the initial rest length.

**ECM degradation.** As protrusions contract, the cell body is pushing against the ECM. In order to allow movement of the cell body, the cell degrades the ECM by proteolytic enzymes like matrix metalloproteinases. Wolf *et al.* demonstrated that proteolysis of collagen fibers does not take place at the protrusion tip, but rather at the cell body where sterically impeding fibers are targeted [34, 35]. Therefore, solid ECM particles within 1.9 μm (slightly shorter than the ECM particle distance of 2 μm) from the cell boundary that is not part of a protrusion ($d_{\mathrm{degr,cell}}$) and with a solid hydrostatic pressure above a threshold pressure $p_{\mathrm{th,degr}}$ are degraded with a chosen degradation rate $r_{\mathrm{degr,cell}}$, representing degradation of the sterically impeding ECM. In order to prevent numerical instability, solid ECM particles within 1.4 μm from the protrusion and with a solid hydrostatic pressure above $p_{\mathrm{th,degr}}$ are degraded with a chosen degradation rate $r_{\mathrm{degr,prot}}$. ECM degradation at the protrusion tip is only allowed during protrusion growth.

**Adhesion disassembly.** Protrusions can contract and pull the cell body until the adhesion disassembles. Here, the adhesion can disassemble with a force-dependent rate $r_{\mathrm{off}}$. Stricker *et al.* observed that inhibition of myosin II-activity reduces the lifetime of mature adhesions, but only at almost complete loss of cellular tension [30]. On the other hand the adhesion is assumed to rupture at high load. The force-dependent adhesion disassembly rate, scaled by the maturation factor, is implemented as:

$$r_{\mathrm{off}} = \begin{cases} r_{\mathrm{off,min}} + r_{\mathrm{off,0}} e^{-\dfrac{\zeta_{\mathrm{diss}} \|\boldsymbol{F}^{\mathrm{adh}}\|}{f_{\mathrm{mat}}}} & \text{for } \|\boldsymbol{F}^{\mathrm{adh}}\| < f_{\mathrm{rup}} f_{\mathrm{mat}} F_{\mathrm{am}} \\ 1 \times 10^6 & \text{for } \|\boldsymbol{F}^{\mathrm{adh}}\| \geq f_{\mathrm{rup}} f_{\mathrm{mat}} F_{\mathrm{am}} \end{cases} . \tag{7}$$

with $r_{\mathrm{off,min}}$ a minimal disassembly rate at normal contractile load, $r_{\mathrm{off,0}}$ an increase in disassembly rate at zero load, $f_{\mathrm{rup}}$ a parameter that defines how much force the adhesion can carry with respect to its own contractile strength before mechanical rupture and $\zeta_{\mathrm{diss}}$ a parameter that regulates the increase in disassembly rate for low adhesion force (see Fig 4A). Adhesion rupture is implemented with a large disassembly rate of $1\times10^6$. The force-dependent adhesion disassembly rate results in an average lifetime $\tau_{\mathrm{adh}} = \dfrac{1}{r_{\mathrm{off}}}$ (see Fig 4B) and an exponential

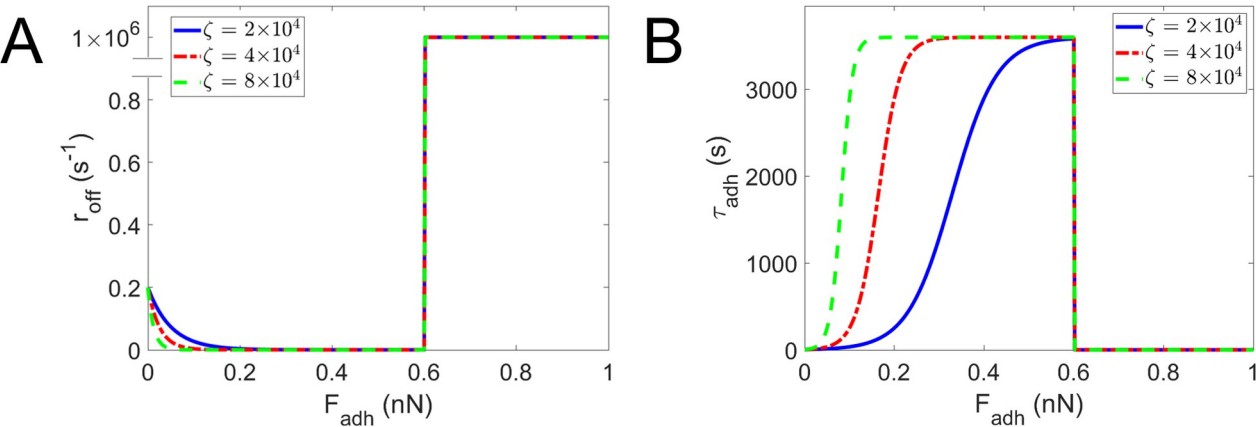

**Fig 4. Force-dependent adhesion disassembly.** (A) Force-dependent adhesion disassembly rate ($r_{off}$) and (B) accompanying average adhesion lifetime ($\tau_{adh}$) as function of adhesion force ($F^{adh}$) for example maturation factor $f_{mat} = 0.5$ and baseline parameter values: reference actomyosin contractile force $F_{am} = 1.2$ nN, minimal disassembly rate $r_{off,min} = 2.778 \times 10^{-4}$ s$^{-1}$, increase in disassembly rate at zero force $r_{off,0} = 0.2$ s$^{-1}$ and force-dependent disassembly rate parameter $\zeta_{diss} = 2 \times 10^4$, $4 \times 10^4$ and $8 \times 10^4$ (see Eq 7 and surrounding text for parameter meanings).

lifetime probability density function:

$$P(t_{adh} > t) = e^{\frac{-t}{\tau_{adh}}} \quad \text{for } t \geq 0, \tag{8}$$

with $t_{adh}$ the lifetime of a single adhesion. Both the average adhesion lifetime $\tau_{adh}$ and the lifetime of a single adhesion $t_{adh}$ represent the lifetime excluding the time for maturation during which adhesions cannot disassemble in the model. When the adhesion disassembles the protrusion continues to contract for $T_{finish} = 400$ s after which $f_{mat}$ is reset to 0.05 (which represents low contractile force in the cell body) and the protrusion particles are available again to form a new protrusion.

## Model implementation

All simulations in this manuscript are performed using the C++ particle-based software called Mpacts (http://dem-research-group.com). The time step, which appears to be limited by the stiff adhesion spring (required for accurate ECM probing in the maturation phase), is set to 0.4 s. The explicit Euler method is used to integrate the density, stress and position of particles in each time step.

## Results

### Stochastic variability in protrusion lifetime regulates cell migration

A parameter study is performed to investigate the effect of protrusion dynamics and the ECM stiffness on cell migration. Multiple sets of cell migration simulations are performed in which a single parameter is varied, while the other parameter values are fixed to values shown in Table 1. In these simulations the cell is embedded in a viscoelastic ECM and migrates for 6 hours. As protrusion initiation and adhesion disassembly are modeled as a stochastic process, 12 simulations are run for each parameter value in a simulation set. In each set simulations are run for 5 different parameter values, resulting in a total of 60 simulations per set. As different protrusions should mature similarly in a homogeneous ECM and should thus become equally strong, they are not expected to be strong enough to rupture adhesions of other protrusions.

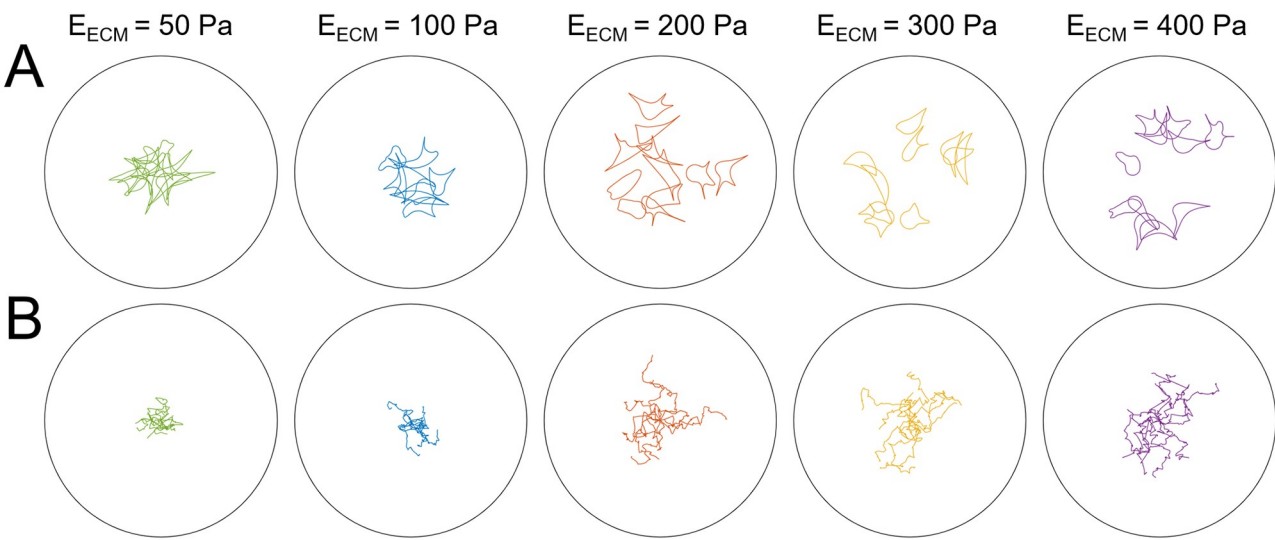

**Fig 5. Cell shapes and pathways for different ECM stiffnesses.** (A) Cell shapes after 6 hours of migration and (B) cell migration paths for cell migration through an ECM with ECM stiffness of 50, 100, 200, 300 and 400 Pa (left to right). The black circle indicates the ECM boundary with a radius of 150 μm. For each ECM stiffness 12 simulations were run.

Therefore, the adhesion rupture parameter $f_{\mathrm{rupt}}$ (see Eq 7) is set very high to prevent the occasional rupture of an adhesion by the contractile force of its own protrusion.

First, the effect of the ECM is investigated by varying the ECM stiffness $E_{\mathrm{ECM}}$ in a range of 50–400 Pa. Cells shapes after 6 hours of migration and cell migration paths are shown in Fig 5A and 5B. Migration of the cell through a degradable ECM at various time points of a simulation is shown in Fig 6 and videos of cell migration in a 100 Pa and 400 Pa ECM are shown in supplementary material (S1–S4 Videos). For ECM stiffness values of 50–200 Pa an increase in ECM stiffness results in a higher average absolute migration velocity ($v_{\mathrm{migr,abs}}$, cell position after 6 hours minus initial cell position, divided by 6 hours), a higher total number of protrusions in 6 hours (#$prot$), a lower average protrusion lifetime ($\tau_{\mathrm{prot}}$, excluding the time for protrusion growth $T_{\mathrm{prot}}$ and maturation $T_{\mathrm{mat}}$) (see Fig 7) and a slightly shorter average protrusion length $l_{\mathrm{prot}}$. Besides, correlation between simulation readouts shows that there is a significant correlation between absolute migration velocity and protrusion length, total number of protrusions and protrusion lifetime (see S2 Fig). The absolute migration velocity rather than the migration velocity (off the cell's center of mass) along the cell path is analyzed here because the former velocity is affected less by protrusion dynamics. Due to the 2D representation of 3D cell migration, the area of a protrusion relative to the cell body area is significant. Therefore, protrusions that retract immediately after being formed do not displace the cell body, but result in a significant migration velocity of the cell's center of mass along the cell path. Cell migration along the cell path could therefore be overestimated and be determined by the number of protrusions rather than the efficiency of protrusions to cause a net displacement of the cell body. The mean absolute migration velocity per stiffness ranges from 3.7–10.2 μm/hr. Experimentally measured 3D cell migration velocities along the cell path, as reported in literature for various ECM and cell types, range from 0–60 μm/hr [10, 35, 40–42]. Since migration velocities along the cell path are higher than absolute migration velocities, the absolute migration velocities measured here are realistic for 3D cell migration. The observed increase in migration velocity with higher ECM stiffness can be explained from the model implementations. As the ECM stiffness increases, protrusions mature more (*i.e.* reach a higher $f_{\mathrm{mat}}$) and

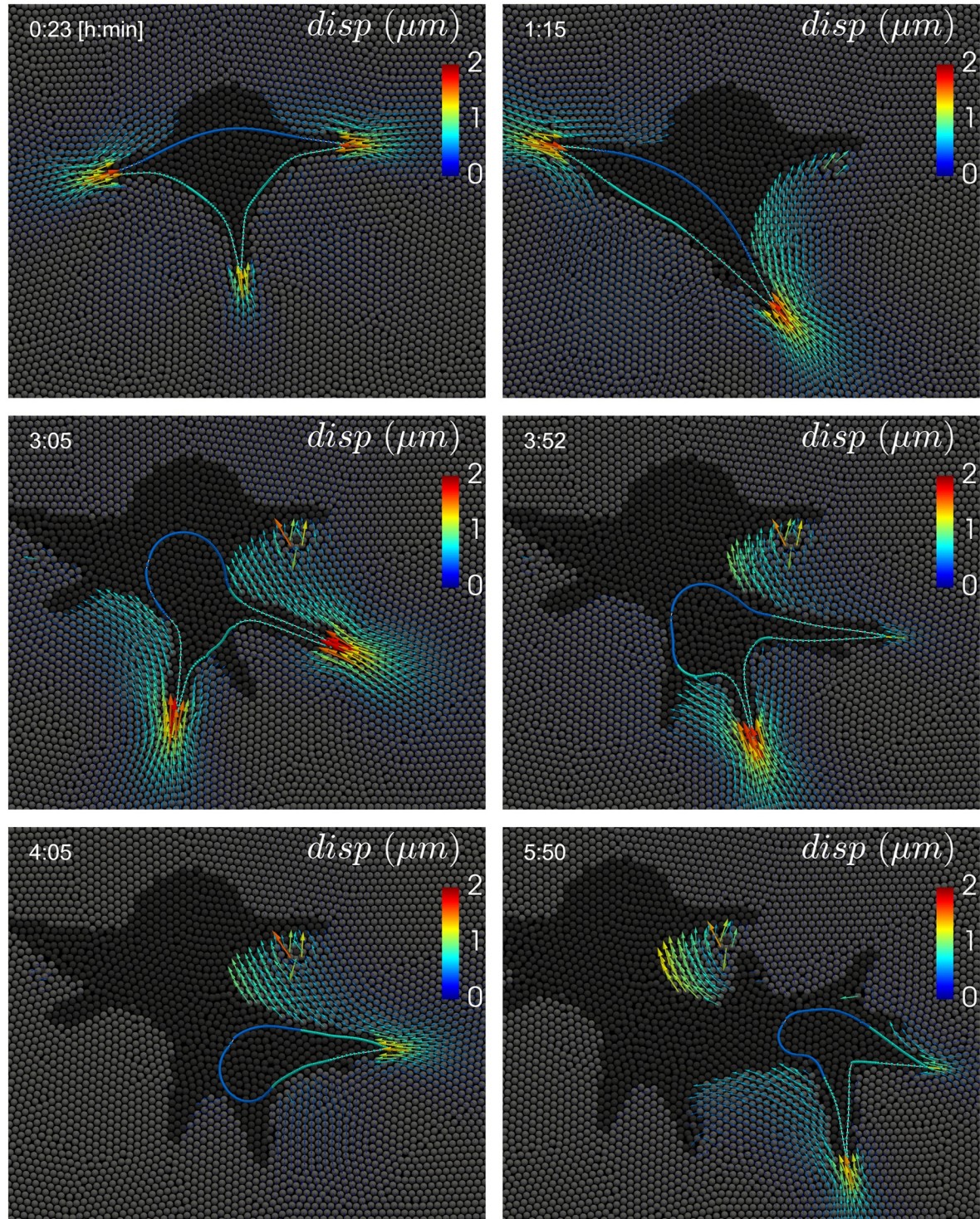

**Fig 6. Simulation result of cell migration through a degradable viscoelastic ECM.** The cell is polarized, with the front colored in turquoise and the rear in blue. The solid ECM ($E = 400$ Pa) is represented by gray particles and the degraded ECM by black particles. Arrows represent the displacement in the solid ECM with respect to the initial particle positions. The cell forms multiple protrusions and creates a tunnel by degrading the ECM. A video of this simulation can be found in S3 Video.

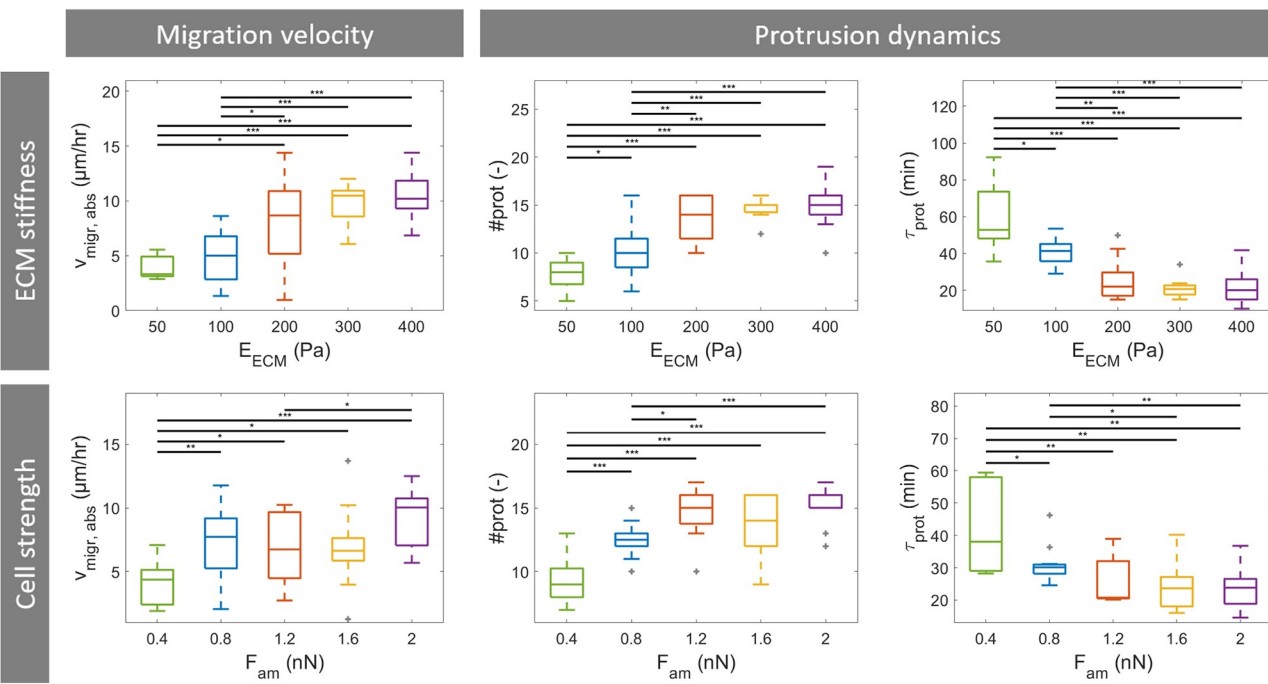

**Fig 7. Results of a cell migration model parameter study for ECM stiffness and cell strength.** Box plots of average absolute cell migration velocity ($v_{\text{migr,abs}}$), total number of protrusions (#$prot$) and protrusion lifetime ($\tau_{\text{prot}}$) as function of 2 different model parameters: ECM stiffness ($E_{\text{ECM}}$, first row) and cell strength (reference actomyosin contractile force $F_{\text{am}}$, second row). For each parameter 5 different parameter values were evaluated and for each parameter value 12 simulations were run. Horizontal bars indicate a significant difference between outcome parameter values of two groups of 12 simulations, calculated by means of the unpaired two-sample t-test. Statistical significance: * $p<0.05$, ** $p<0.01$, *** $p<0.005$. Outliers shown in gray.

become stronger. A stronger contraction of the protrusion actin cortex leads to faster pulling of the cell body towards the adhesion and therefore earlier relaxation of the cortex when it reaches its minimal length. This means that the force on the adhesion is reduced faster, which increases the adhesion disassembly rate and thereby decreases the average protrusion lifetime. Automatically, as the average protrusion lifetime drops, a new protrusion can be formed earlier, resulting in a higher total number of protrusions during 6 hours of migration. This explains why cells migrate further away from their initial position in an ECM with a higher stiffness. An increase in ECM stiffness in the range of 200–400 Pa does not show significant changes in cell migration velocity, total number of protrusions and average protrusion lifetime (see Fig 7), although the protrusion maturation still increases significantly for these ECM stiffness values. This can be explained from the fact that the cell usually forms 2 or 3 protrusions that can stabilize each other by pulling on the cell body in different directions. As long as a protrusion cannot pull the cell body towards the adhesion, the protrusion maintains its contractile force, which preserves a high adhesion force and thus a low adhesion disassembly rate in the order of $r_{\text{off,min}}$. Only when one of the adhesions of competing protrusions disassemble, pulling of the cell body towards the competing protrusion (and its shortening) can start, which results in a decrease in force on the remaining adhesion and thus an increase in adhesion disassembly rate. At high protrusion strength (high $f_{\text{mat}}$) the gain in protrusion contraction speed with further increase in protrusion strength appears to be negligible compared to the long protrusion strength independent phases in which protrusions are formed or compete with and stabilize each other. Therefore, an increase in protrusion strength at high ECM stiffness is found to not further increase cell migration velocity.

Next, the effect of cell strength is investigated by varying the reference actomyosin contractile force $F_{am}$ in a range of 0.4–2 nN. This change in cell strength is not expected to affect the number of maturation steps for a protrusion, but will change the contractile force of both the protrusions and the cell body (see Eq 6). An increase in cell strength results in similar effects as seen for an increase in ECM stiffness (see Fig 7). Again, a significant increase in average absolute migration velocity and total number of protrusions and a decrease in average protrusion lifetime are observed for an increase in cell strength at low reference actomyosin contractile force values. At high cell strength values no significant further changes are observed.

As a change in average absolute migration velocity in these two simulation sets is accompanied by a higher total number of protrusions, a lower average protrusion lifetime and, in the case of varying ECM stiffness, a lower average protrusion length, the influence of these three cellular properties on cell migration is investigated. First, the total number of protrusions is varied by changing the protrusion initiation rate $r_{prot}$ at each cell particle ranging from $1\times10^{-5}$ $s^{-1}$ to $1\times10^{-4}$ $s^{-1}$. With 235 particles per cell, it follows that an attempt of protrusion initiation is made at on average every 42.5–425 s. However, as a protrusion can be initiated only when the particle is at the front of the polarized cell and if there are enough cell boundary particles available that are not already part of a protrusion, the effective protrusion formation rate is lower. The average total number of protrusions in 6 hours of migration ranges from 8 to 15 protrusions for the range of protrusion initiation rate values (see Fig 8). However, the average absolute migration velocity is not affected by the protrusion initiation rate (see Fig 8). Moreover, no significant correlation is observed between absolute migration velocity and total number of protrusions (see S2 Fig). The reason for this is that the increase in protrusion number results in an increase in time during which multiple protrusions are competing with and stabilize each other. This can be seen from the fact that the protrusion lifetime is unaffected by or even slightly increased with an increase in total number of protrusions (no significant correlation). An increase in protrusion strength in the previous simulation sets resulted in a larger number of protrusions because protrusions shortened faster by pulling the cell body in the adhesion direction. Therefore, these protrusions lived shorter as the adhesion force decreased faster, which thus resulted in more effective migration.

Second, the average protrusion lifetime is varied by changing the adhesion disassembly rate at normal load $r_{off,min}$ ranging from $1.85\times10^{-4}$ $s^{-1}$ to $16.67\times10^{-4}$ $s^{-1}$. These values are selected such that the expected average lifetimes at normal load in the different simulations are 10, 20, 40, 60 and 90 minutes. It can be observed that the average protrusion lifetime decreases with an increase in $r_{off,min}$. As a result, the average total number of protrusions during 6 hours of cell migration increases. However, as was seen for protrusion initiation rate, the average absolute migration velocity is not affected by the adhesion disassembly rate at normal force (see Fig 8) and there is no significant correlation between absolute migration velocity and average protrusion lifetime (see S2 Fig). The reason for this is that a reduction in protrusion lifetime also reduces the protrusion contraction efficiency. Adhesions disassemble earlier and thus protrusions pull the cell body less far in the adhesion direction. In contrast, for scenarios that are accompanied by an increase in cell strength protrusions live shorter because adhesion force decreases faster when the cell body is pulled in the adhesion direction. Therefore, cell body displacement by a single protrusion remains the same despite a shorter protrusion lifetime and an increase in total number of protrusion can therefore result in more migration.

Finally, the protrusion length is varied by changing the protrusion growth time $T_{prot}$ ranging from 200–600 s. It can be observed that an increase in protrusion growth time results in an increase in average protrusion length (see Fig 8). The protrusion length does not double if the protrusion growth time is doubled because the protrusion tip becomes thinner and thus provides less membrane area for the polymerizing actin to push against as the protrusion grows.

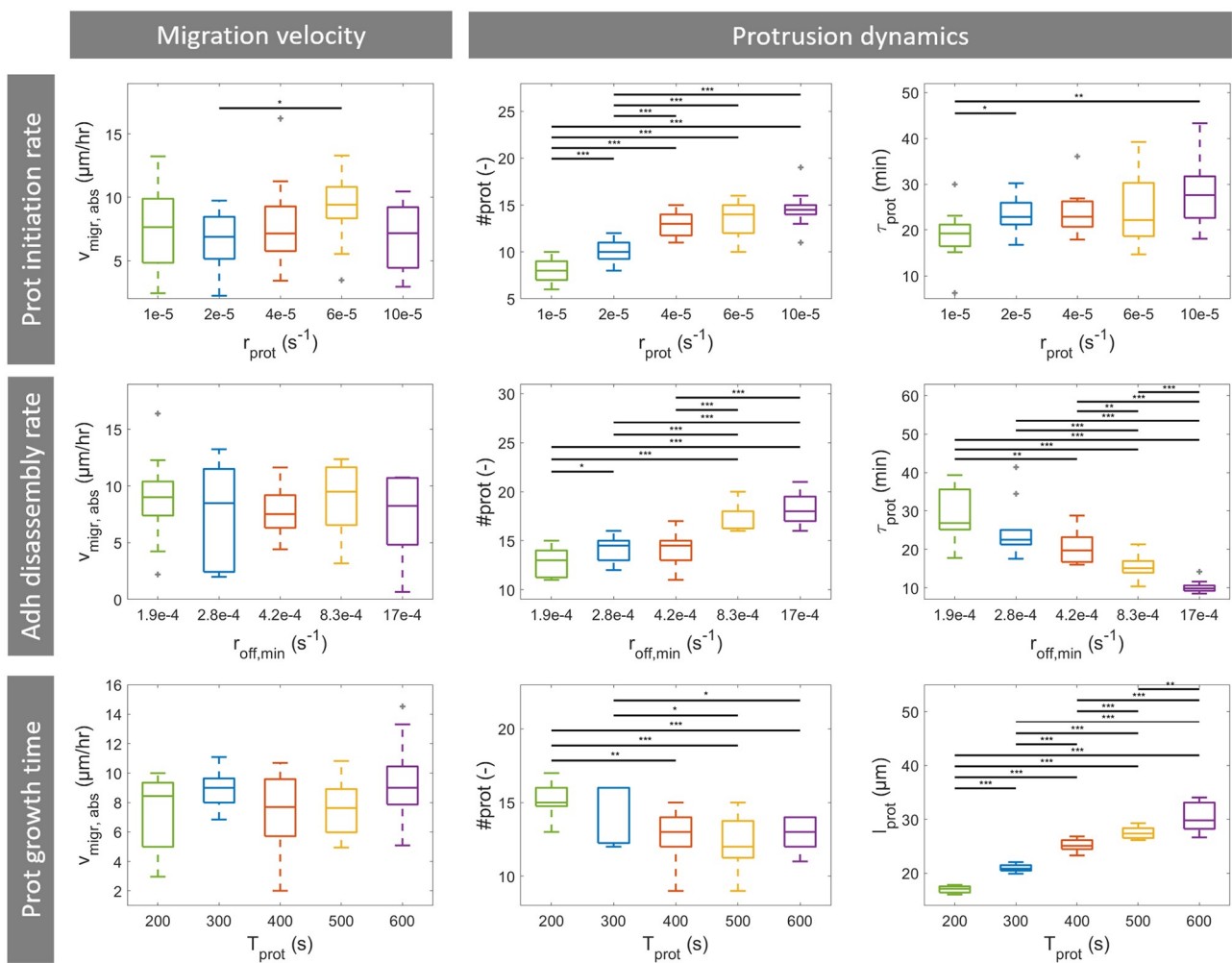

**Fig 8. Results of a cell migration model parameter study.** Box plots of average absolute cell migration velocity ($v_{migr,abs}$), total number of protrusions (*#prot*), protrusion lifetime ($\tau_{prot}$) and protrusion length ($l_{prot}$) as function of 3 different model parameters: total number of protrusions (protrusion initiation rate $r_{prot}$, first row), average protrusion lifetime (adhesion disassembly rate at normal load $r_{off,min}$, second row) and average protrusion length (protrusion growth time $T_{prot}$, third row). For each parameter 5 different parameter values were evaluated and for each parameter value 12 simulations were run. Horizontal bars indicate a significant difference between outcome parameter values of two groups of 12 simulations, calculated by means of the unpaired two-sample t-test. Statistical significance: * $p<0.05$, ** $p<0.01$, *** $p<0.005$. Outliers shown in gray.

An increase in average protrusion length does not affect the average absolute migration velocity (see Fig 8 and S2 Fig). It can be seen that an increase in protrusion length is accompanied by a slight decrease in average total number of protrusions and a slight increase in the average protrusion lifetime (see S2 Fig). This can be explained by the fact that for a longer protrusion it takes more time to form the protrusion and pull the cell body in the adhesion direction. So an increase in protrusion length can result in more cell body displacement per protrusion, but due to the lower total number of protrusions the cell does not migrate further. Besides, the increase in protrusion lifetime also increases the time during which multiple protrusions compete with and stabilize each other, which can also slow down migration.

The results described above illustrate that ECM stiffness and cell strength are important regulators of cell migration. Strong cells on the one hand can pull themselves quickly in an adhesion direction and can therefore migrate by making many short living protrusions. Weaker cells on the other hand need more time to pull themselves in an adhesion direction

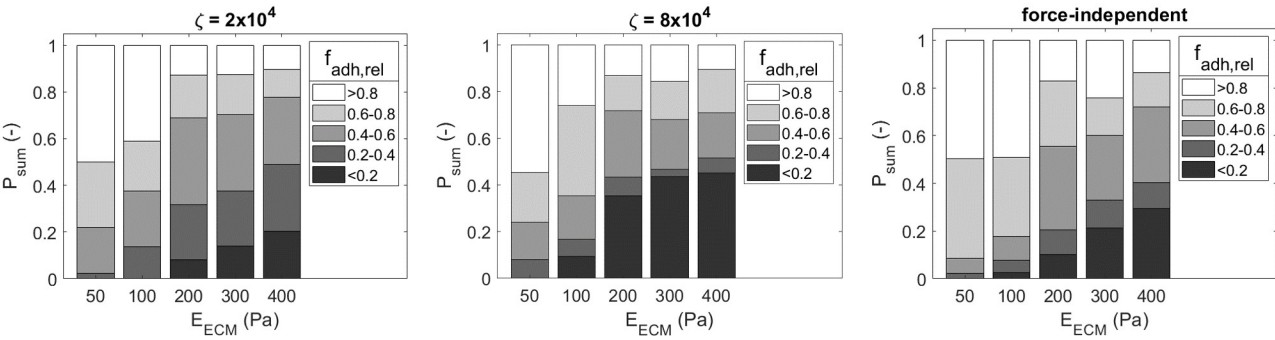

**Fig 9. Distribution of relative force at adhesion disassembly for cell migration with force-dependent and force-independent adhesion disassembly.** Distribution of relative force ($f_{\text{adh,rel}}$) at adhesion disassembly as function of ECM stiffness ($E_{\text{ECM}}$) for force-dependent ($\zeta_{\text{diss}} = 2\times10^4$ and $\zeta_{\text{diss}} = 8\times10^4$, see Eq 7) and force-independent adhesion disassembly rate.

and therefore migrate by making less and more long-living protrusions. At high adhesion force, even though the average protrusion lifetime equals 60 minutes ($\frac{1}{r_{\text{off,min}}}$), due to the exponential lifetime probability density function (see Eq 8) the majority of protrusions will live shorter than the average lifetime while only a few protrusions might live (much) longer. This large stochastic variability in protrusion lifetime can explain the increase in absolute migration velocity for stronger cells as protrusions of stronger cells contract faster and thus a larger percentage of protrusions will have completely pulled the cell towards the adhesion at the time of disassembly. This can be seen from the distribution of the relative contractile force at the time of adhesion disassembly ($f_{\text{adh,rel}}$), which is calculated as the adhesion force at disassembly divided by the contractile force after maturation:

$$f_{\text{adh,rel}} = \frac{\|\boldsymbol{F}^{\text{adh}}\|}{f_{\text{mat}F_{\text{am}}}}. \tag{9}$$

This ratio is an indirect indicator of protrusion efficiency, which is the amount of cell body displacement per protrusion, as its value decreases only when the protrusion shortens and the cell body is pulled towards the adhesion. An increase in ECM stiffness results in a higher percentage of adhesions that disassemble at low $f_{\text{adh,rel}}$ and thus more protrusions have pulled the cell in the adhesion direction by the time their adhesion disassembles, therefore making them more effective (see Fig 9, left).

$r_{\text{off}}$ is implemented with a low rate for high adhesion force and a high rate for low adhesion force (See Fig 4A). In this way adhesions of protrusions that start contracting are less likely to disassemble than adhesions of protrusions that have already shortened and pulled the cell body towards the adhesion. In order to investigate the effect of this force-dependent disassembly rate two additional simulation sets are performed. In a first set $\zeta_{\text{diss}}$, the parameter that regulates the range of forces at which adhesions are stabilized (see Eq 7), is increased to $8\times10^4$ to increase this stabilization range (see Fig 4). It can be seen in Fig 9 (center) that a larger percentage of adhesions disassemble at low relative adhesion force for all ECM stiffnesses. This results in a significant decrease in total number of protrusions and increase in protrusion lifetime for an ECM stiffness of 200 Pa and higher, but not in a significant change in average absolute migration velocity. Therefore, the increase in adhesion lifetime for a larger adhesion force range does not increase cell migration velocity, but makes migration more efficient by reducing the total number of protrusions and increasing the cell displacement per protrusion contraction, which is expected to be energetically favorable.

In a second set a force-independent adhesion disassembly rate ($r_{\text{off}}$) with a value of $5.56 \times 10^{-4}$ $s^{-1}$ is prescribed, which gives an average adhesion lifetime of 30 minutes. This results in an increase in disassembly rate at high adhesion force and a decrease in disassembly rate at low adhesion force compared to a force-dependent rate with a value for $\zeta_{\text{diss}}$ of $2 \times 10^4$ or $8 \times 10^4$. Compared to simulations with $\zeta_{\text{diss}} = 2 \times 10^4$ the number of protrusions that disassemble at $f_{\text{adh,rel}} < 0.4$ is reduced for cells in a low stiffness ECM (see Fig 9, right) and thus a lower percentage of protrusions effectively displace the cell body. For high ECM stiffness the number of protrusions that disassemble at $f_{\text{adh,rel}} < 0.2$ is increased, which means that these protrusions have pulled the cell body in the adhesion direction. However, since the adhesion disassembly rate does not increase after the adhesion force has decreased, some of these protrusions might also exist for too long as their adhesions do not disassemble directly after pulling the cell body in the adhesion direction. When both simulations with $\zeta_{\text{diss}} = 2 \times 10^4$ and $\zeta_{\text{diss}} = 8 \times 10^4$ are compared to simulations with force-independent adhesion disassembly rate no significant differences in average absolute migration velocity are observed for all ECM stiffnesses (see Fig 10B and 10C). However, for $\zeta_{\text{diss}} = 2 \times 10^4$ the average total number of protrusions is lower and the average protrusion lifetime is higher for 50 and 100 Pa ECM stiffness, while the inverse is observed for 300 Pa ECM stiffness. For $\zeta_{\text{diss}} = 8 \times 10^4$ the average total number of protrusions is lower for 50, 100, 200 and 400 Pa and the average protrusion lifetime is higher for 50, 100 and 200 Pa.

In summary, changes in ECM stiffness and cell strength affect cell migration and are accompanied by changes in protrusion dynamics, in particular protrusion number and lifetime. However, directly changing these protrusion dynamics does not affect cell migration. Results show that a force-dependent adhesion disassembly rate does not increase cell migration velocity. Therefore, the model suggests that a stochastic variability in protrusion lifetime (exponential adhesion lifetime probability density function, see Eq 8) is already enough to optimize migration for cells in ECMs with various stiffnesses. Instead of affecting the migration velocity, a force-dependent adhesion disassembly rate reduces the number of protrusions required to obtain a similar migration velocity and therefore makes migration more efficient.

## Optimal number of simultaneous protrusions depends on ECM anisotropy

Fraley *et al.* revealed that focal adhesion proteins can modulate cell migration through a 3D matrix by regulating protrusion dynamics [28]. Effective cell migration could be assured by establishing a low number of protrusions. They hypothesized that the optimal number of major protrusions at a time should lie between zero, for which cells would not be able to move, and not more than two, above which cells would not be able to move persistently as protrusions would pull in too many directions simultaneously. In order to test if an optimal number of protrusions exists for effective cell migration, simulations are performed in which the formation of 1, 2, 3 or 4 simultaneous protrusions is enforced. At every time step a cell particle is selected to initiate a new protrusion. This protrusion is allowed to form only if the number of existing protrusions is lower than the prescribed number of protrusions $n_{\text{prot}}$. Cell particles at the rear can be selected in order to permit formation of 3 or 4 protrusions. However, they are selected with a 100 times lower rate than particles at the front in order to preserve cell polarity. In order to prevent excessive cell area growth, protrusions can be initiated only if the cell area is smaller than or equal to twice the initial cell area. Besides, the protrusion force is scaled by the current cell area $A_{\text{cell}}$ over the initial cell area $A_{0,\text{cell}}$, mimicking

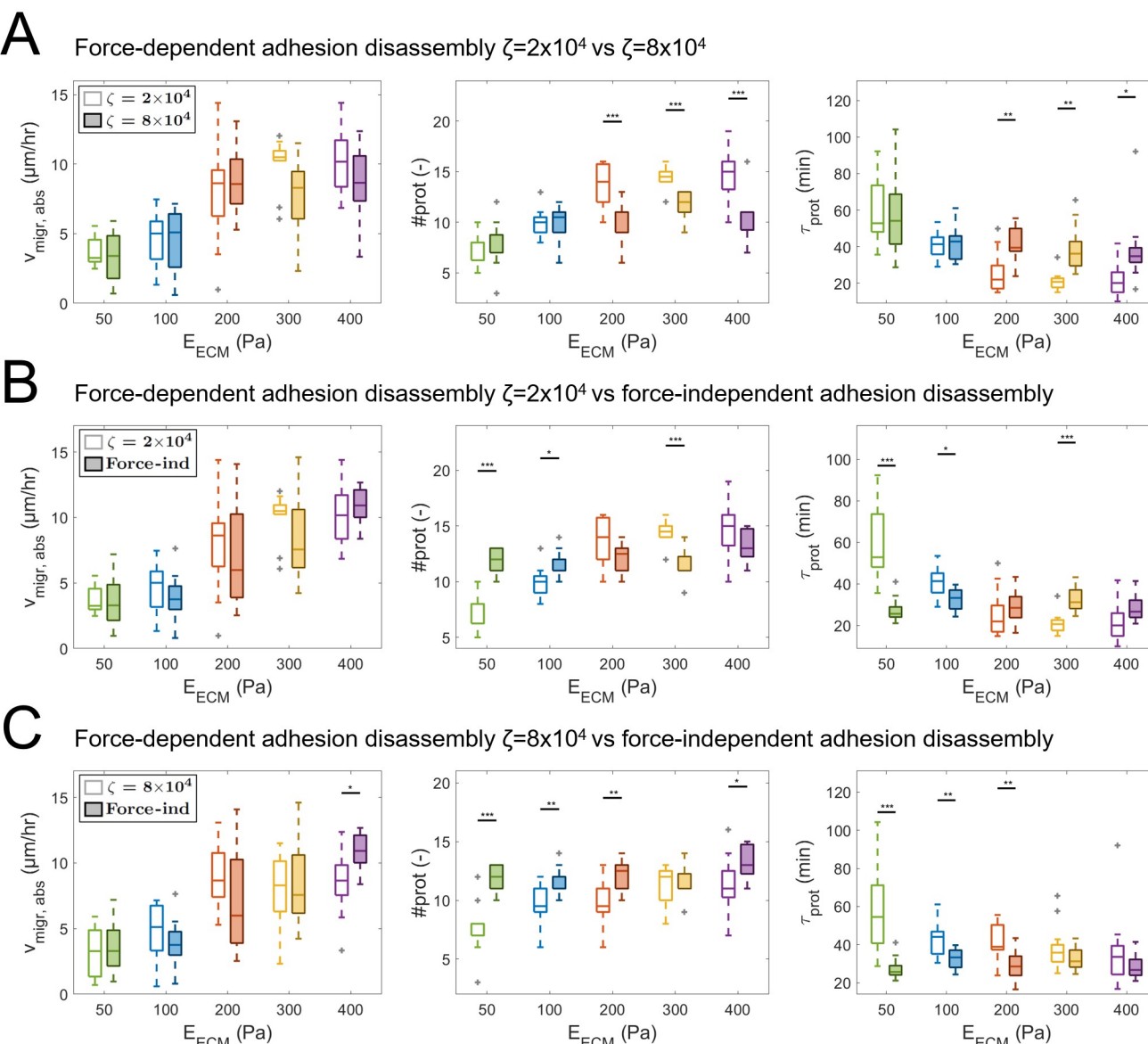

**Fig 10. Simulation results for cell migration with force-dependent and force-independent adhesion disassembly.** Simulation results for force-dependent ($\zeta_{\text{diss}} = 2 \times 10^4$ and $\zeta_{\text{diss}} = 8 \times 10^4$, see Eq 7) and force-independent adhesion disassembly rate. (A–C) Comparison of box plots of average absolute cell migration velocity ($v_{\text{migr,abs}}$), total number of protrusions (#prot) and protrusion lifetime ($\tau_{\text{prot}}$) as function of force-dependent and force-independent adhesion disassembly. Horizontal bars indicate a significant difference between outcome parameter values of two groups of 12 simulations, calculated by means of the unpaired two-sample t-test. Statistical significance: * $p < 0.05$, ** $p < 0.01$, *** $p < 0.005$. Outliers shown in gray.

the reduction in available actin when the cell increases in size:

$$\|\boldsymbol{F}^{\text{prot}}\| = F_{\text{prot}} \frac{A_{0,\text{cell}}}{A_{\text{cell}}} \qquad \text{for } A_{\text{cell}} \geq A_{0,\text{cell}} \tag{10}$$

First, cells are placed in a homogeneous viscoelastic ECM with Young's modulus $E = 200$ Pa and are allowed to migrate for 6 hours. Final cell shapes, cell migration paths and results of cell migration analysis are shown in Fig 11. It can be observed that the average absolute migration velocity ($v_{\text{migr,abs}}$, cell position after 6 hours minus initial cell position, divided by 6 hours)

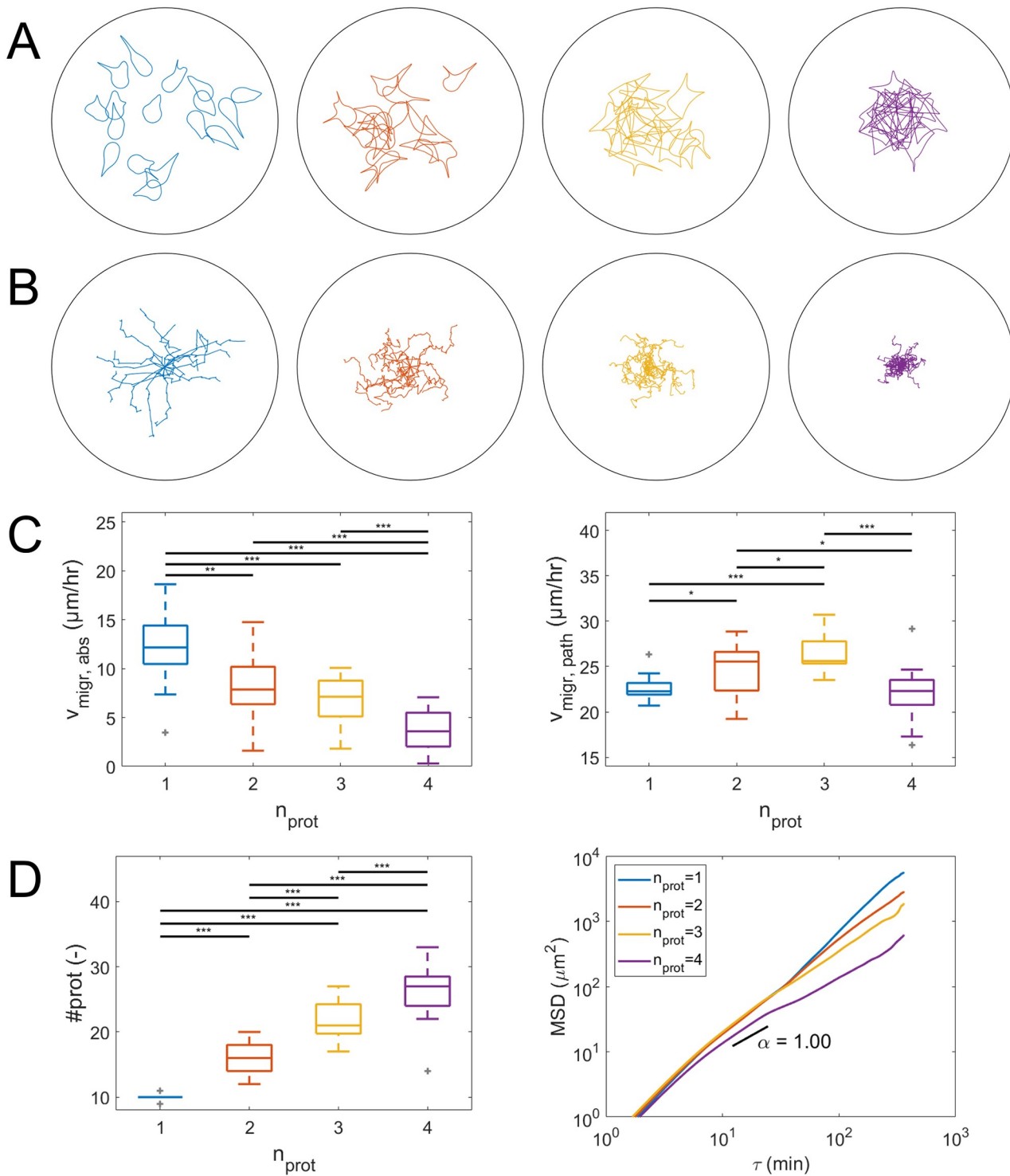

**Fig 11. Simulation results for cell migration through an isotropic viscoelastic ECM (without any nonlinear elastic springs).** Results for cells with 1 (blue, n = 17), 2 (red, n = 17), 3 (yellow, n = 17) or 4 (magenta, n = 17) simultaneous protrusions. (A) Cell shapes after 6 hours of migration. The black circle indicates the ECM boundary with a radius of 150 μm. (B) Cell paths representing the cell center of mass displacement during 6 hours. (C) Absolute cell migration velocity ($v_{\mathrm{migr,abs}}$, cell position after 6 hours minus initial cell position, divided by 6 hours) and migration velocity along cell path ($v_{\mathrm{migr,path}}$, total path length divided by 6 hours). (D) Total number of protrusions (#*prot*) as function of the prescribed number of simultaneous protrusions ($n_{prot}$) and MSD as function of time lag $\tau$ (log-log plot), where $\alpha = 1$ represents the slope of the MSD for a random walk. Horizontal bars indicate a significant difference between outcome parameter values of two groups of 12 simulations, calculated by means of the unpaired two-sample t-test. Statistical significance: * p<0.05, ** p<0.01, *** p<0.005. Outliers shown in gray.

decreases with increase in number of simultaneous protrusions, with 12.1 μm/hr for cells with one protrusion and 3.7 μm/hr for cells with 4 protrusions. At the same time, cells with only one protrusion clearly form the least total number of protrusion during 6 hours of migration, indicating that the formation of multiple protrusions strongly reduces absolute migration velocity. The total number of protrusions in 6 hours increases with the prescribed number of simultaneous protrusions, which shows that inhibition of protrusion initiation at large cell area does not prevent cell migration. Migration along the cell path is highest for a cell with 3 protrusions, with an average migration velocity along the cell path ($v_{\text{migr,path}}$, total path length divided by 6 hours) of 26.6 μm/hr, and lowest for cells with either 1 or 4 protrusions, for which the velocity along the cell path is around 22 μm/hr. The cell displacement generated per protrusion is clearly highest for 1 simultaneous protrusion, which is expected as multiple protrusions can counteract each other and thereby prevent cell body displacement. These results together indicate that cells with less protrusions migrate more direct and efficient (higher cell body displacement per protrusion) as can be seen from both the cell migration paths (straighter paths) and the mean squared displacements (MSDs) as function of time lag $\tau$ (steeper slope, with $\text{MSD}(\tau) = \langle [\boldsymbol{x}(t + \tau) - \boldsymbol{x}(t)]^2 \rangle$ and time $t$). The slope of the MSD of cells with 4 protrusions is close to 1, which represents a random walk, while the higher slope for cells with less protrusions indicates that they follow a straighter path.

Next, cells are placed in an anisotropic, uniaxial fibrillar ECM (with nonlinear elastic springs in one direction and a lower Young's modulus of 10 Pa for the nonfibrillar ECM component, see Fig 2) in order to investigate the effect of the number of simultaneous protrusions on cell migration through an anisotropic fibrillar ECM. Final cell shapes, cell migration paths and results of cell migration analysis are shown in Fig 12. It can be seen from the migration paths that cells migrate preferentially along the fiber direction. Since no preferential protrusion growth direction due to *e.g.* contact guidance is implemented, this demonstrates that mechanosensing by protrusions is enough to guide cell migration. Further, it can also be observed that cells with one protrusion do not migrate further in 6 hours than cells with 2 protrusions, which now have the highest average absolute migration velocity of 10.7 μm/hr (although not significantly different from cells with one protrusion), and only slightly (but significantly) further than cells with 3 or 4 protrusions. Cells with one protrusion are also significantly slower along their cell path (14.6 μm/hr) than cells with multiple protrusions, with the highest average migration along the cell path of 21.2 μm/hr for cells with 2 protrusions. Compared to migration in a homogeneous viscoelastic ECM, the average total number of protrusions in 6 hours for cells with one protrusion has dropped from 9.9 to 5.6. This happens because protrusions that try to protrude in a direction that is not aligned with the fiber (spring) direction sense a very weak ECM and do not mature much. As a result these protrusions are weaker and take longer to shorten and pull the cell body towards the corresponding adhesion. Therefore, it takes longer for the adhesion force to decrease and for the adhesion disassembly rate to increase, which explains the higher average protrusion lifetime. Cells with multiple protrusions are more likely to form at least one protrusion in the fiber direction that will mature more and become stronger, allowing it to rupture the adhesions of neighboring weaker and thus slower contracting protrusions. This results in a more efficient way of migration in which strong protrusions rapidly displace the cell body while weaker protrusions are quickly retracted due to adhesion rupturing. Therefore, migration has become more directed and faster for cells with multiple protrusions for migration in an anisotropic, uniaxial fibrillar ECM compared to a homogeneous ECM. This can also be observed from an increase in the slope of the MSD in Fig 12D compared to Fig 11D. An example of how this competition between protrusions can result in migration along the fiber direction is shown in Fig 13 for a cell with 2 simultaneous protrusions. This figure shows the Von Mises stress distribution in the uniaxial fibrillar ECM

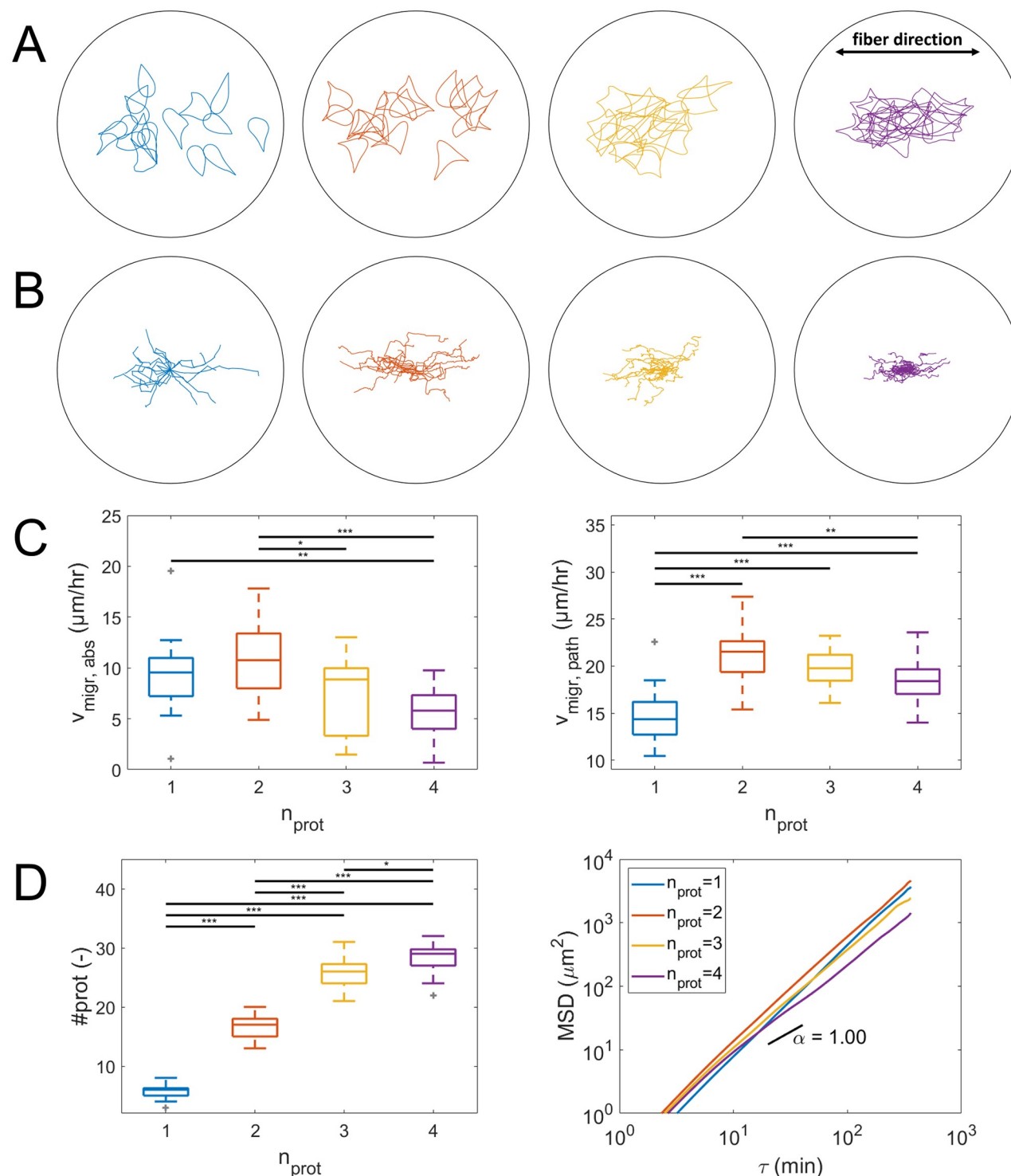

**Fig 12. Simulation results for cell migration through an anisotropic, uniaxial fibrillar ECM (nonlinear elastic springs in horizontal direction).** Results for cells with 1 (blue, n = 17), 2 (red, n = 17), 3 (yellow, n = 17) or 4 (magenta, n = 19) simultaneous protrusions. (A) Cell shapes after 6 hours of migration. The black circle indicates the ECM boundary with a radius of 150 μm. (B) Cell paths representing the cell center of mass displacement during 6 hours. (C) Absolute cell migration velocity ($v_{migr,abs}$, cell position after 6 hours minus initial cell position, divided by 6 hours) and migration velocity along cell path ($v_{migr,path}$, total path length divided by 6 hours). (D) Total number of protrusions (#prot) as function of the prescribed number of simultaneous protrusions ($n_{prot}$) and MSD as function of time lag $\tau$ (log-log plot), where $\alpha$ = 1 represents the slope of the MSD for a random walk. Horizontal bars indicate a significant difference between outcome parameter values of two groups of 12 simulations, calculated by means of the unpaired two-sample t-test. Statistical significance: * p<0.05, ** p<0.01, *** p<0.005. Outliers shown in gray.

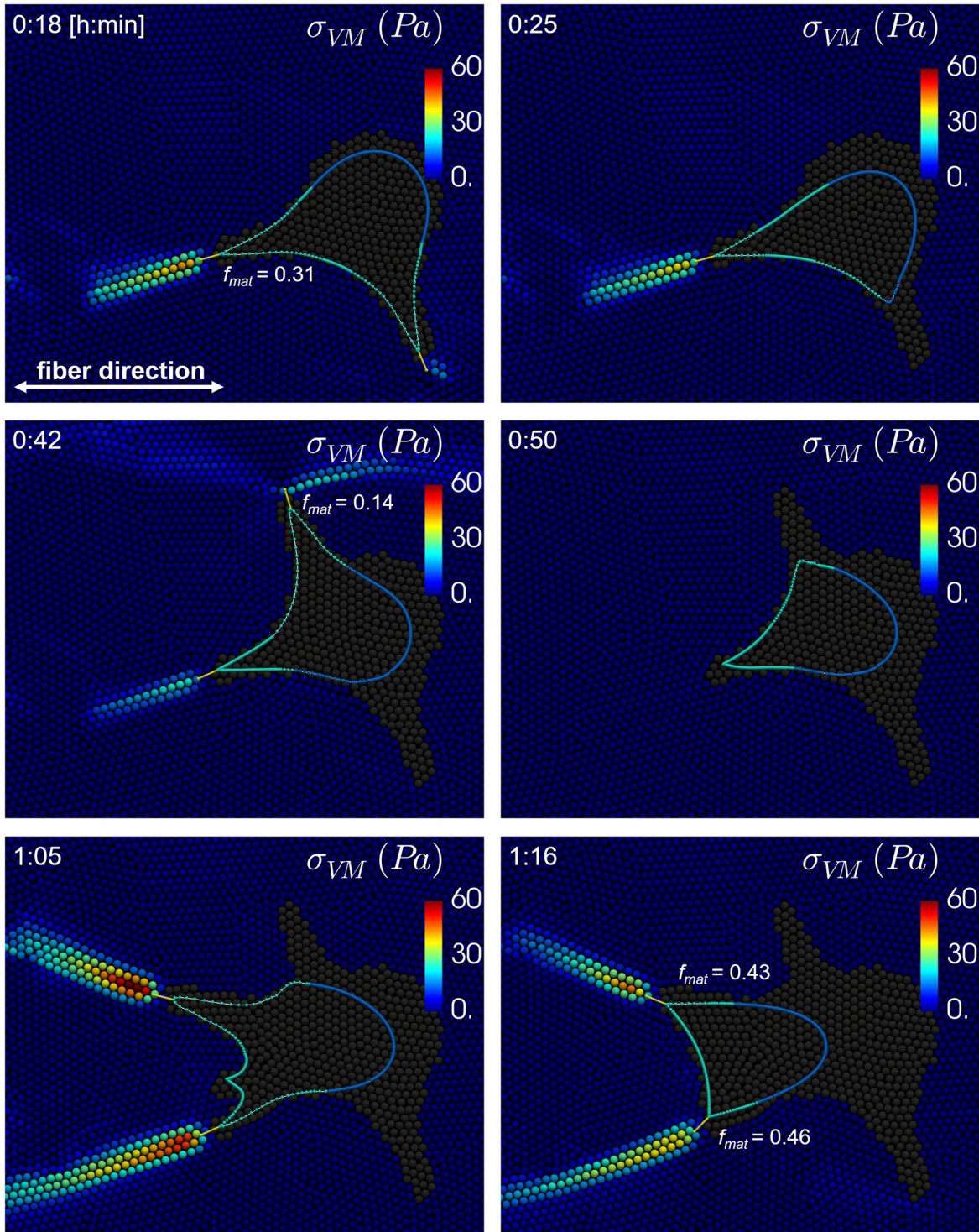

**Fig 13. Competition between protrusions for 3D migration through an anisotropic, uniaxial fibrillar ECM for a cell with 2 simultaneous protrusions.** The cell is polarized, with the front colored in turquoise and the rear in blue. The von Mises stress in the fibrillar part of the ECM is shown for solid ECM particles, while the degraded ECM is represented by black particles. Protrusions formed in the direction perpendicular to the fiber direction are weaker (lower $f_{mat}$) and therefore quickly retracted due to rupture of the corresponding adhesions, allowing the cell to polarize and migrate along the fiber direction.

during migration. Since the stress in the nonfibrillar part of the ECM model is negligible compared to the stress in the fibrillar part, the Von Mises stress is calculated only for the fibrillar part. The Von Mises stress is calculated from the Cauchy stress tensor. This Cauchy stress tensor is calculated for each solid ECM particle, based on a derivation by Cioroianu *et al.* for an elastic spring network [43], as:

$$\boldsymbol{\sigma}_i^{c,\alpha\beta} = \sum_{j \in \mathcal{S}\backslash i} \frac{1}{V_i \det(\mathcal{F}_{ij})} \|\boldsymbol{F}_{ij}^{\text{fib}}\| \left( \frac{\boldsymbol{x}_{ij}^{\alpha} \boldsymbol{x}_{ij}^{\beta}}{\|\boldsymbol{x}_{ij}\|} \right), \tag{11}$$

written in Einstein notation with respect to the coordinate indices $\alpha$ and $\beta$, with $\boldsymbol{\sigma}^c$ the Cauchy stress tensor, $V = \dfrac{m}{\rho}$ the particle volume and $\mathcal{F}$ the deformation gradient tensor for the spring between particles $i$ and $j$.

Together, these results confirm the hypothesis of Fraley *et al.* that the number of protrusions should ideally lie between 0 and not more than 2 protrusions [28]. Migration is most efficient with one protrusion in a homogeneous ECM. In an anisotropic ECM with a preferred fiber direction, mechanosensing by multiple protrusions improves migration efficiency, with most efficient migration for 2 simultaneous protrusions. More than 2 protrusions increases the probability of forming opposing protrusions that hinder cell body displacement.

## Discussion

In this paper a computational model was developed to investigate the role of actin protrusion dynamics and ECM properties on 3D cell migration. Cell migration was modeled with a hybrid approach combining an agent-based mechanical cell model and a meshless Lagrangian particle-based degradable viscoelastic ECM model. The cell model captures the main subcellular processes required for migration, *i.e.* membrane protrusion, cell-ECM adhesion, actomyosin contraction and ECM degradation. By probing the local ECM stiffness and applying a corresponding contractile force, migration is adapted to the ECM. The ECM model describes the mechanics of either an isotropic, viscoelastic ECM or an anisotropic, uniaxial fibrillar ECM.

First, it was shown that changes in ECM stiffness and cell strength affect cell migration and are accompanied by changes in number, lifetime and (only slightly) length of protrusions. Directly varying the parameter values that govern protrusion dynamics did not result in changes in cell migration. As a force-dependent adhesion lifetime did not affect cell migration velocity, the model suggested that a stochastic variability in adhesion lifetime was already enough to optimize migration of cells in ECMs with different stiffnesses. Instead of affecting the migration velocity, a force-dependent adhesion disassembly rate reduced the number of protrusions required to obtain a similar migration velocity and therefore made migration more efficient. Second, the hypothesis of Fraley *et al.* that the optimal number of simultaneous protrusions should lie between 0 and 2 was confirmed and further refined [28]. The formation of maximal 1 protrusion proved to be most efficient for migration in a homogeneous ECM. For cells in an anisotropic ECM with preferred fiber direction the formation of 2 protrusions proved to be most efficient as competition between mechanosensing protrusions was required for orienting the cell front.

Our current understanding of the role and regulation of protrusion dynamics for 3D cell migration is limited. Fraley *et al.* discovered that 3D cell migration speed (along the cell path) is correlated mainly to the number of protrusions per time, while protrusion lifetime and length are not significantly correlated to migration speed. In our model, varying protrusion number, lifetime and length did not affect the average absolute migration velocity. However, a

change in absolute migration velocity as a result of a change in cell strength was accompanied by changes in protrusion number, lifetime and length. At the same time, no significant difference in migration velocity was observed for cells with both force-dependent and force-independent adhesion disassembly, while protrusion number and lifetime were significantly affected. Our simulation results demonstrate the complex interplay between cell migration and protrusion dynamics, and the fact that correlations between protrusion features (such as number and lifetime) and cell migration velocity do not necessarily imply a causal relation. Altogether, our results show that cell migration speed in our model is regulated mainly by cell strength and ECM stiffness (due to mechanosensing), while force-dependent adhesion disassembly is required to optimize migration efficiency (cell body displacement per protrusion), which is expected to be energetically favorable. While our computational model enables to isolate (perturb) specific subcellular processes and assess their direct effect on cell migration (without perturbing other processes), it is very difficult to almost impossible to do that in an experiment, which demonstrates the added value of the model.

While the model captures the main features of 3D cell migration, some processes were simplified or neglected. First, the model is a 2D representation of 3D migration. This makes it more difficult to compare migration paths and velocities with 3D cell migration experiments. Besides, this affects the hindrance experienced by the cell model due to the surrounding ECM because for a 2D model the cell is only in contact with the ECM in the considered plane. Second, protrusion initiation and growth are modeled independent of the surrounding ECM and cell mechanics. Fischer *et al.* revealed that pseudopodial branching of endothelial cells is inhibited by ECM stiffness and myosin II activity and that local depletion of myosin II precedes branch formation [29]. They also showed that this regulation of branch formation by myosin II contraction results in more directed and faster cell migration. Elliott *et al.* discovered that myosin II contractility minimizes cellular branching by minimizing the local curvature of the cell surface [33]. They hypothesized that this could favor the formation of protrusions along the elongation direction of a cell over protrusions oriented perpendicular to the elongation direction, thereby increasing persistence of migration direction. Cells are also known to use filopodia to sense local chemical and mechanical cues, which allows them to regulate the formation of protrusions and direct cell migration [44]. Although protrusion initiation in our model is not regulated by local myosin II contraction, ECM stiffness or chemical cues, the formation of an excessive number of protrusions is prevented by allowing protrusions to form only at the front of the cell. In most simulations this limits the number of protrusions that can exist at the same time to 3 protrusions. When the cell was allowed to form protrusions also at the rear in order to obtain 4 simultaneous protrusions, the migration velocity decreased and migration was less directed. This shows that the formation of an excessive number of protrusions hinders cell migration because the protrusions pull in opposing directions, which is prevented by allowing protrusion to form only at the cell front.

Next, cells are known to use the local ECM fiber orientation to guide protrusion growth and therefore enhance migration efficiency by increasing directional persistence [42, 45, 46]. The model presented here did not implement contact guidance by fibers as protrusion growth was not made dependent on collagen fiber direction (which was captured by springs between ECM particles). Although such implementation might result in even more directed cell migration, the results here indicate that contact guidance is not necessary for a cell in order to follow a preferred collagen fiber direction. Competition between multiple protrusions that probe the stiffness of the local ECM in multiple directions and, through actomyosin contraction, rupture the adhesions of weak protrusions (namely those protrusions that form in softer ECM directions) is enough to explain directed cell migration. The current ECM model could be used to investigate the effect of structured ECM architectures on 3D cell migration. For example

stiffness gradients, rigid obstacles, specific ECM geometries and mechanical anisotropy by favoring fiber direction can already be implemented easily. Currently, the main limitation of the model is that springs in the ECM model do not represent individual collagen fibers, but are a coarse-grained representation of the nonlinear mechanical behavior. Therefore, the cell does not attach to and pull on individual collagen fibers and the ECM does not present a porous structure for the cell to migrate through, which This makes it challenging to distinguish between an ordered and disordered fiber structure.

Ehrbar *et al.* showed that 3D cell migration in a PEG hydrogel is reduced by an increase in gel stiffness [40]. The same effect was observed by Wolf *et al.* for migration through a collagen gel, while Lang *el al.* and Fraley *et al.* observed biphasic responses for migration through a collagen gel with either highest or lowest migration velocity at intermediate ECM stiffnesses [35, 42, 47]. These observations are different from our observation that cell migration increases with ECM stiffness. An explanation for this is that an increase in ECM stiffness is accompanied by an increase in cross-linking density, which hinders cell migration. In order to migrate through a densely cross-linked gel more ECM degradation is required. In our model ECM degradation in front of a growing protrusion is fast in order to allow protrusions to form. Degradation of ECM particles close to the cell body is slower, but still quick compared to a real hydrogel, as slow or no degradation occasionally resulted in numerical instabilities when a solid ECM particle was pushed through the cell boundary. Therefore, the cell model experiences less hindrance from the ECM than what might occur in reality and migration velocity is determined mainly by contractile strength of the cell, which increases with ECM stiffness (due to the mechanotransduction mechanism depicted in Fig 3F). Mason *et al.* developed a method to tune the stiffness of a collagen scaffold without changing the collagen density and observed an increase in endothelial cell spreading and outgrowth with an increase in collagen stiffness, which is in agreement with our observations [48]. In the future, it would be interesting to investigate if the observed responses to ECM stiffness and anisotropy still hold when a more realistic hindrance of cell migration would be captured. The limited hindrance of cell migration by the ECM in our model, compared to experiments, can also be derived from the differences in simulated versus experimentally observed cell paths. Wu *et al.* demonstrated that an anisotropic persistent random walk model is required to describe experimentally observed 3D cell migration, where the anisotropic part captures the preferential reorientation of cells in microchannels created by ECM degradation [41]. In our model, deflection of protrusion growth based on the local ECM (as described in S4 Text) increases the likeliness of protrusion growth into already degraded ECM areas, favoring to some extent these directions for cell migration. At the same time, as in our model protrusion initiation does not depend on the local ECM density, cell migration is not restricted to existing microchannels in the ECM.

The current model could be adapted and extended to investigate various cases of both single cell migration and collective migration. An important condition would be that the studied migration mode should be based on similar mechanisms as those described here, namely protrusion formation, cell-ECM adhesion, actomyosin contraction and ECM degradation. Therefore, amoeboid migration, which is observed for example for neutrophils during inflammation, would be less suitable to investigate with our model, mainly because ECM degradation is required for migration due to the lack of micrometer-sized pores in the ECM model. In order to investigate migration for a specific cell type and extracellular environment, tuning of the model based on experiments will be important. Properties such as protrusion dynamics, adhesion lifetimes and ECM degradation can be observed through optical microscopy, while 3D traction force microscopy is an interesting technique to tune the contractile strength of the cell model. Depending on the cell type, regulation of subcellular processes by external chemical signals, resulting in chemotaxis, might also be an indispensable for understanding migration.

Intracellular signaling, regulated by external chemical signals, could be modeled by means of partial differential equations at the cell domain and would be a logical future extension of the model. Finally, collective cell migration could be modeled with the current model by adding cell-cell adhesion and intercellular signaling. It would be particularly interesting to investigate the effect of tunnel formation in the ECM on migration of strands of cells as seen in for example angiogenesis and collective cancer cell metastasis.

In conclusion, we have proposed a new computational model of 3D cell migration that captures the mechanics and dynamics underlying cell migration. To the best of our knowledge, this is the first model that combines a mechanical deformable cell model, which migrates by extending and contracting protrusions that probe the local ECM stiffness, with a deformable and degradable ECM model. By investigating the effect of protrusion dynamics, cell strength and ECM mechanics we have demonstrated that this model is able to provide new insights in the role and regulation of protrusion dynamics in 3D cell migration and the way cell migration is adapted to the local ECM. Therefore, we believe that this model can be a valuable contribution to increase the understanding of 3D cell migration mechanisms. In the future, this model be could extended further by regulating the initiation and growth direction of protrusions based on sensing of mechanical and chemical cues by filopodia and chemical signaling pathways inside the cell. Besides, the addition of a deformable nucleus is necessary if cell migration is hindered due to slower ECM degradation, forcing the cell to deform strongly in order to move through narrow openings in the ECM. Finally, a tighter coupling with experiments is required. Ideally these experiments should allow to vary or tune processes that are captured by model parameters, such as ECM stiffness, ECM degradability and protrusion dynamics, to further increase our understanding of 3D cell migration.

## Supporting information

**S1 Text. Deformable cell model.**
(PDF)

**S2 Text. SPH implementation of the ECM model.**
(PDF)

**S3 Text. ECM degradation and cell-ECM boundary conditions.**
(PDF)

**S4 Text. Protrusion deflection.**
(PDF)

**S5 Text. Cortex curvature-dependent actomyosin contraction.**
(PDF)

**S1 Fig. Cell polarization.**
(PDF)

**S2 Fig. Simulation readout correlation.**
(PDF)

**S1 Video. ECM displacement field for cell migration through ECM with stiffness of 100 Pa.**
(MP4)

**S2 Video. ECM Von Mises stress distribution for cell migration through ECM with stiffness of 100 Pa.**
(MP4)

**S3 Video. ECM displacement field for cell migration through ECM with stiffness of 400 Pa.**
(MP4)

**S4 Video. ECM Von Mises stress distribution for cell migration through ECM with stiffness of 400 Pa.**
(MP4)

## Acknowledgments

The authors thank Simon Vanmaercke and Tim Odenthal for technical support in designing the computational algorithms for the numerical simulations. This work utilized the high-performance computing resources of the Vlaams Supercomputer Centrum in Flanders, Belgium (https://www.vscentrum.be/).

## Author Contributions

**Conceptualization:** Tommy Heck, Diego A. Vargas, Bart Smeets, Herman Ramon, Paul Van Liedekerke, Hans Van Oosterwyck.

**Data curation:** Tommy Heck.

**Formal analysis:** Tommy Heck.

**Funding acquisition:** Tommy Heck, Hans Van Oosterwyck.

**Investigation:** Tommy Heck, Diego A. Vargas.

**Methodology:** Tommy Heck, Diego A. Vargas, Bart Smeets.

**Project administration:** Tommy Heck, Paul Van Liedekerke, Hans Van Oosterwyck.

**Resources:** Herman Ramon, Hans Van Oosterwyck.

**Software:** Tommy Heck, Diego A. Vargas, Bart Smeets, Paul Van Liedekerke.

**Supervision:** Paul Van Liedekerke, Hans Van Oosterwyck.

**Validation:** Tommy Heck.

**Visualization:** Tommy Heck.

**Writing – original draft:** Tommy Heck.

**Writing – review & editing:** Tommy Heck, Bart Smeets, Paul Van Liedekerke, Hans Van Oosterwyck.

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
