## [Decision Letter · Decision Letter 0]

2 Sep 2019

Dear Dr Heck,

Thank you very much for submitting your manuscript 'The role of actin protrusion dynamics in cell migration through a degradable viscoelastic extracellular matrix: Insights from a computational model' for review by PLOS Computational Biology. Your manuscript has been fully evaluated by the PLOS Computational Biology editorial team and in this case also by independent peer reviewers. The reviewers appreciated the attention to an important problem, but raised some substantial concerns about the manuscript as it currently stands. While your manuscript cannot be accepted in its present form, we are willing to consider a revised version in which the issues raised by the reviewers have been adequately addressed. We cannot, of course, promise publication at that time.

In particular, the reviewers comment that there is little to no validation of your predictions with experimental results. Although experimental validation  is not required for publication, we do ask that you evaluate the plausibility of the experimental predictions against published observations. The reviewers suggests several possibilities for this. Also, while the model is presented repeatedly as a model for 3D cell migration, the model is two-dimensional. Please clarify that the model is a 2D representation of a 3D situation. Please address these, as well the other comments of the referees in a revised version of your manuscript.

Sincerely,

Roeland M.H. Merks, Ph.D

Associate Editor

PLOS Computational Biology

Mark Alber

Deputy Editor

PLOS Computational Biology

[LINK]

Reviewer's Responses to Questions

**Comments to the Authors:**

Reviewer #1: Review is uploaded as an attachment

Reviewer #2: It was a pleasure to review “The role of actin protrusion dynamics in cell migration through a degradable viscoelastic extracellular matrix: Insights from a computational model” by Heck et al. My background is in the biology of cell migration, so I have reviewed the manuscript from that perspective and limit my comments to the biological aspects of the study and the accessibility of the study to experimental biologists, including the plausibility of the biological assumptions and the significance & novelty of the conclusions. I make no comment on the mathematical or computational aspects of the study.

This manuscript tackles the important question of the regulation of 3D cell migration which is an important element of many physiological and pathological processes including wound healing, the immune response and cancer metastasis. This is a question that is challenging, although not impossible, to address experimentally and so would benefit from a computational model that allows rapid interrogation of hypotheses. The computational model is very well described and clear to biologists. It makes plausible assumptions about the behaviour of cells, cellular protrusions and the ECM and the model that is generated allows for significant and novel conclusions about the behaviour of cells in three dimensional environments to be reached. These conclusions have the ability to be experimentally verified making them useful for cell biologists working in this field.

Overall, I thought this was a well written study that will be of broad interest to cell biologists. Below I make some minor comments that could be addressed to strengthen the manuscript.

1. The model in this study represents a model of generic cell migration and the manuscript may benefit from discussion about how this might model some specific example of cell migration, for example cancer cells during ECM invasion as a part of metastasis or neutrophils responding to chemokines during infection. These can be processes that are very different in their regulation, with different scope for ECM degradation, different requirements for chemotaxis and can result in cells transitioning through different ECM environments. This is also a model of individual cell migration through ECM and I wonder if a similar approach would also model collective cell migration which is often seen during wound healing and some types of cancer metastasis.

2. Throughout the manuscript the authors generally refer to “cell protrusions”. Filopodia are mentioned several times but to me the protrusions as described in the manuscript, particularly their involvement in ECM degradation, may better resemble invadopodia or podosomes (depending on whether the cells in questions are cancer cells or normal cells). Some brief mention or discussion of these invasive types of protrusions would be helpful.

3. There is a lack of experimental verification of the model. There is some discussion of the Fraley paper that supports some of the model conclusions, but other conclusions lack experimental support. I’m not suggesting that experimental support is necessary in this study and indeed I feel it is both extremely challenging to do and well outside the scope of the study but perhaps some discussion of future experimental steps would be helpful.

4. Some of the model parameters shown in Table 1 are well chosen and supported by the literature but some – such as cortex stiffness, actomyosin force, maturation time etc – are not and are described as trial runs or model setup. How were these parameters chosen and how well do they reflect biological reality? While I imagine some of these parameters have not been measured in 3D systems, there may be quantification of them in 2D systems that may at least be a reasonable starting point.

5. Just a comment rather than anything else but the geometry or architecture of the ECM in vivo is often more ordered or structured than that found in many in vitro systems (such as collagen gels) which often have chaotic and disorganised fibre structure. The model in this manuscript is modelling in vitro environments but could easily be adapted to model in vivo ECM environments with different ECM architecture. It would be interesting to see if that changes the behaviour of the cells.

6. A very minor point – on page 12 the sentence “Protrusions grow by weakening the actin cortex and pushing the membrane outwards” initially read as if it was describing bleb formation. I assume – as is made clear on page 13 – that “pushing the membrane outwards” is accomplished by actin filament polymerisation rather than through hydrostatic pressure but this could be made clearer.

Reviewer #3: In this work, authors present an hybrid approach combining an agent-based mechanical cell model and a meshless Lagrangian particle-based degradable viscoelastic ECM model in order to simulate 3D cell migration.

In my opinion, the most interesting contribution of this work is the novel numerical approach that has been proposed. In fact, it is really novel and innovative approach to simulate 3D cell migration by means of this hybrid approach. However, in the case of the model, there are some assumptions and simplifications that should be commented and better justified. Three main assumptions are my main concern in this work:

• In this work, it is assumed that cell polarization is well defined, clearly fixing the front and the rear of the cell. However, the polarization of the cell during migration in 3D is not well defined, being multiple, because many protrusions are competing between them to define their movement.

• The cell nucleus has not been considered in the model and however its role is crucial in 3D migration. I think it is a strong simplification in the model for simulating 3D migration.

• Although it is true that there are not differences from a concept point of view, cell movement is being really modeled as a 2D phenomenon. In my opinion, it is a limitation when you have to interpret results from in-vitro experiments, in which movement is clearly 3D. 

Some other assumptions also require further analysis or justification:

• There are some works that indicate that the main component of protrusions is not actin, but tubulin (Fraley et al, 2012).

• The assumption that the number of contraction steps required to reach a threshold force is highly dependent on the matrix rigidity could be controversial, because in the experimental work of Mitrossilis et al (2009, PNAS), they measured that the time required for achieving the threshold force was independent of this stiffness.

• In this work, it is assumed that cells are able to generate more force in the direction of the protrusions in function of the matrix stiffness. Have you considered that this force can saturate? Please, see again Mitrossilis et al (2009, PNAS).

• It is not clear for me how the model is able to simulate the porosity of the extracellular matrix, above all in the case of fibrillar hydrogels like collagen gels.

• The degradation of the ECM is not clear how has been simulated. In one part of the paper, it is indicated that “ECM particles close to the protrusion tip or the cell body can be degraded with a chosen degradation rate” and in another one is indicated: “Wolf et al demonstrated that proteolysis of collagen fibers does not take place at the protrusion tip, but rather at the cell body where sterically impeding fibers are targeted”. Please, revise carefully the paper to avoid this misunderstanding.

• Adhesions are assumed to rupture at high load, but adhesions also fail at low load. In fact, they are normally simulated by a catch-bond law (see Escribano et al, 2018, BMMB).

Finally, the predicted results about 3D durotaxis are a direct consequence of the considered assumptions. However, there is much debate in the bibliography whether or not 3D durotaxis really exists.

**Have all data underlying the figures and results presented in the manuscript been provided?**

Reviewer #1: None

Reviewer #2: Yes

Reviewer #3: Yes

PLOS authors have the option to publish the peer review history of their article (what does this mean?). If published, this will include your full peer review and any attached files.

Reviewer #1: No

Reviewer #2: No

Reviewer #3: No

---

## [Decision Letter · Decision Letter 1]

28 Nov 2019

Dear Dr Heck,

Thank you very much for submitting your manuscript, 'The role of actin protrusion dynamics in cell migration through a degradable viscoelastic extracellular matrix: Insights from a computational model', to PLOS Computational Biology. As with all papers submitted to the journal, yours was fully evaluated by the PLOS Computational Biology editorial team, and in this case, by independent peer reviewers. The reviewers appreciated the attention to an important topic but identified some aspects of the manuscript that should be improved.

Please address the comments by reviewer 1, in particular point 3. 

We would therefore like to ask you to modify the manuscript according to the review recommendations before we can consider your manuscript for acceptance. Your revisions should address the specific points made by each reviewer and we encourage you to respond to particular issues Please note while forming your response, if your article is accepted, you may have the opportunity to make the peer review history publicly available. The record will include editor decision letters (with reviews) and your responses to reviewer comments. If eligible, we will contact you to opt in or out.raised.

- Supporting Information uploaded as separate files, titled 'Dataset', 'Figure', 'Table', 'Text', 'Protocol', 'Audio', or 'Video'.

We hope to receive your revised manuscript within the next 30 days. If you anticipate any delay in its return, we ask that you let us know the expected resubmission date by email at ploscompbiol@plos.org.

Sincerely,

Roeland M.H. Merks, Ph.D

Associate Editor

PLOS Computational Biology

Mark Alber

Deputy Editor

PLOS Computational Biology

[LINK]

Reviewer's Responses to Questions

**Comments to the Authors:**

Reviewer #1: The review is uploaded as an attachment.

Reviewer #3: I think authors have improved the paper with the review that they have developed.

**Have all data underlying the figures and results presented in the manuscript been provided?**

Reviewer #1: None

Reviewer #3: Yes

PLOS authors have the option to publish the peer review history of their article (what does this mean?). If published, this will include your full peer review and any attached files.

Reviewer #1: No

Reviewer #3: No

---

## [Editor Report · Decision Letter 2]

5 Dec 2019

Dear Dr Heck,

We are pleased to inform you that your manuscript 'The role of actin protrusion dynamics in cell migration through a degradable viscoelastic extracellular matrix: Insights from a computational model' has been provisionally accepted for publication in PLOS Computational Biology.

In the meantime, please log into Editorial Manager at https://www.editorialmanager.com/pcompbiol/, click the "Update My Information" link at the top of the page, and update your user information to ensure an efficient production and billing process.

One of the goals of PLOS is to make science accessible to educators and the public. PLOS staff issue occasional press releases and make early versions of PLOS Computational Biology articles available to science writers and journalists. PLOS staff also collaborate with Communication and Public Information Offices and would be happy to work with the relevant people at your institution or funding agency. If your institution or funding agency is interested in promoting your findings, please ask them to coordinate their releases with PLOS (contact ploscompbiol@plos.org).

Thank you again for supporting Open Access publishing. We look forward to publishing your paper in PLOS Computational Biology.

Sincerely,

Roeland M.H. Merks, Ph.D

Associate Editor

PLOS Computational Biology

Mark Alber

Deputy Editor

PLOS Computational Biology

---

## [Editor Report · Acceptance letter]

6 Jan 2020

PCOMPBIOL-D-19-01125R2 

The role of actin protrusion dynamics in cell migration through a degradable viscoelastic extracellular matrix: Insights from a computational model

Dear Dr Heck,

I am pleased to inform you that your manuscript has been formally accepted for publication in PLOS Computational Biology. Your manuscript is now with our production department and you will be notified of the publication date in due course.

With kind regards,

Matt Lyles
